# DDMI: Domain-Agnostic Latent Diffusion Models for Synthesizing High-Quality Implicit Neural Representations

**Dogyun Park, Sihyeon Kim, Sojin Lee, Hyunwoo J. Kim**[*]
Department of Computer Science
Korea University
Seoul, South Korea
{gg933,sh_bs15,sojin_lee,hyunwoojkim}@korea.ac.kr

## Abstract

Recent studies have introduced a new class of generative models for synthesizing implicit neural representations (INRs) that capture arbitrary continuous signals in various domains. These models opened the door for domain-agnostic generative models, but they often fail to achieve high-quality generation. We observed that the existing methods generate the weights of neural networks to parameterize INRs and evaluate the network with fixed positional embeddings (PEs). Arguably, this architecture limits the expressive power of generative models and results in low-quality INR generation. To address this limitation, we propose **D**omain-agnostic Latent **D**iffusion **M**odel for **I**NRs (DDMI) that generates adaptive positional embeddings instead of neural networks' weights. Specifically, we develop a Discrete-to-continuous space Variational AutoEncoder (D2C-VAE) that seamlessly connects discrete data and continuous signal functions in the shared latent space. Additionally, we introduce a novel conditioning mechanism for evaluating INRs with the hierarchically decomposed PEs to further enhance expressive power. Extensive experiments across four modalities, *e.g.*, 2D images, 3D shapes, Neural Radiance Fields, and videos, with seven benchmark datasets, demonstrate the versatility of DDMI and its superior performance compared to the existing INR generative models. Code is available at https://github.com/mlvlab/DDMI.

## 1 Introduction

Implicit neural representation (INR) is a popular approach for representing arbitrary signals as a continuous function parameterized by a neural network. INRs provide great *flexibility* and *expressivity* even with a simple neural network like a small multi-layer perceptron (MLP). INRs are virtually domain-agnostic representations that can be applied to a wide range of signals domains, such as image (Müller et al., 2022; Tancik et al., 2020; Sitzmann et al., 2020), shape/scene model (Mescheder et al., 2019; Peng et al., 2020; Park et al., 2019), video reconstruction (Nam et al., 2022b; Chen et al., 2022), and novel view synthesis (Mildenhall et al., 2021; Martin-Brualla et al., 2021; Park et al., 2021; Barron et al., 2021). Also, INR enables the continuous representation of signals at arbitrary scales and complex geometries. For instance, given an INR of an image, applying zoom-in/out or sampling an arbitrary-resolution image is readily achievable, leading to superior performance in super-resolution (Chen et al., 2021; Xu et al., 2021). Lastly, INR represents signals with high quality leveraging the recent advancements in parametric positional embedding (PE) (Müller et al., 2022; Cao & Johnson, 2023).

Recent research has expanded its attention to INR generative models using Normalizing Flows (Dupont et al., 2022a), GANs (Chen & Zhang, 2019; Skorokhodov et al., 2021; Anokhin et al., 2021), and Diffusion Models (Dupont et al., 2022a; Zhuang et al., 2023). Especially, Dupont et al. (2022a); Zhuang et al. (2023); Du et al. (2021); Dupont et al. (2022b) have focused on developing a generic framework that can be applied across different signal domains. This is primarily

---
[*]Corresponding author.

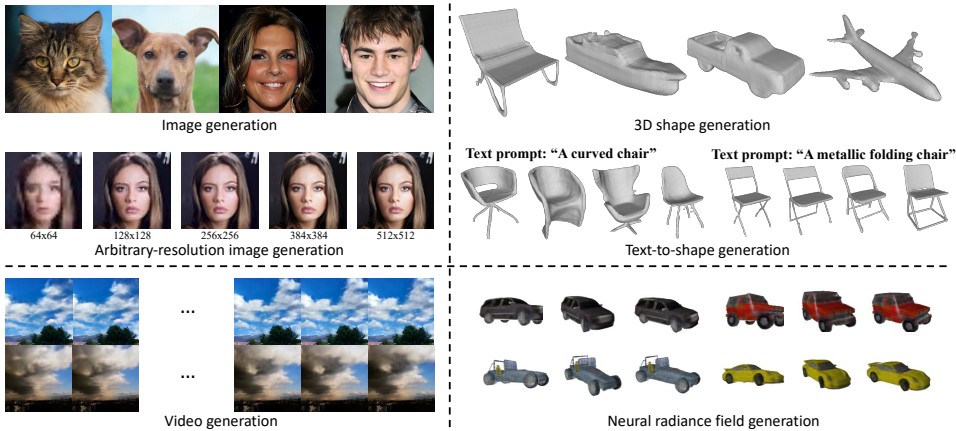

Figure 1: **Generation results of DDMI**. Our DDMI generates high-quality samples across *four distinct domains* including *image, shape, video,* and *Neural Radiance Fields*. DDMI also shows remarkable results for applications like arbitrary-scale image generation or text-to-shape generation.

accomplished by modeling the distribution of INR 'weights' by GAN (Dupont et al., 2022b), latent diffusion model (Dupont et al., 2022a), or latent interpolation (Du et al., 2021). However, these models often exhibit limitations in achieving high-quality results when dealing with large and complex datasets. Arguably, this is mainly due to their reliance on generating weights for an INR function with fixed PEs. This places a substantial burden on function weights to capture diverse details in multiple signals, whereas the careful designs of PE (Müller et al., 2022; Chan et al., 2022; Cao & Johnson, 2023) have demonstrated greater efficiency and effectiveness in representing signals.

Therefore, in this paper, we propose **D**omain-agnostic Latent **D**iffusion **M**odel for **I**NRs (**DDMI**) that generates adaptive positional embeddings instead of neural networks' weights (see Fig. 3 for conceptual comparison). Specifically, we introduce a Discrete-to-continuous space Variational AutoEncoder (D2C-VAE) framework with an encoder that maps discrete data into the latent space and a decoder that maps the latent space to continuous function space. D2C-VAE generates *basis fields* using the decoder network conditioned on a latent variable. This means we define sample-specific basis functions for generating adaptive PEs, shifting the primary expressive power from MLP to PE. Additionally, we propose two modules that further enhance the expressive capacity of INR: 1) Hierarchically Decomposed Basis Fields (HDBFs): we *decompose* the basis fields into *multiple scales* to better account for the multi-scale nature of signals. 2) Coarse-to-Fine Conditioning (CFC): we introduce a novel conditioning method, where the multi-scale PEs from HDBFs are *progressively conditioned* on MLP in a coarse-to-fine manner. Based on D2C-VAE, we train the latent diffusion model on the shared latent space (see Fig. 2 for the overall framework). Ultimately, our model can generate high-quality continuous functions across a wide range of signal domains (see Fig. 1). To summarize, our contributions are as follows:

- We introduce the Domain-agnostic Latent Diffusion Model for INRs (DDMI), a generative model synthesizing high-quality INRs across various signal domains.
- We define a Discrete to Continuous space Variational AutoEncoder (D2C-VAE) that generates adaptive PEs and learns the shared latent space to connect the discrete data space and the continuous function space.
- We propose Hierarchically-Decomposed Basis Fields (HDBFs) and Coarse-to-Fine Conditioning (CFC) to enhance the expressive power.
- Extensive experiments across four modalities and seven benchmark datasets demonstrate the versatility of DDMI. The proposed method significantly outperforms the existing INR generative models, demonstrating the efficacy of our proposed methods.

## 2 RELATED WORKS

**INR-based generative models.** Several works have explored the use of INR in generative modeling to leverage its continuous nature and expressivity. Especially the INR generative models are

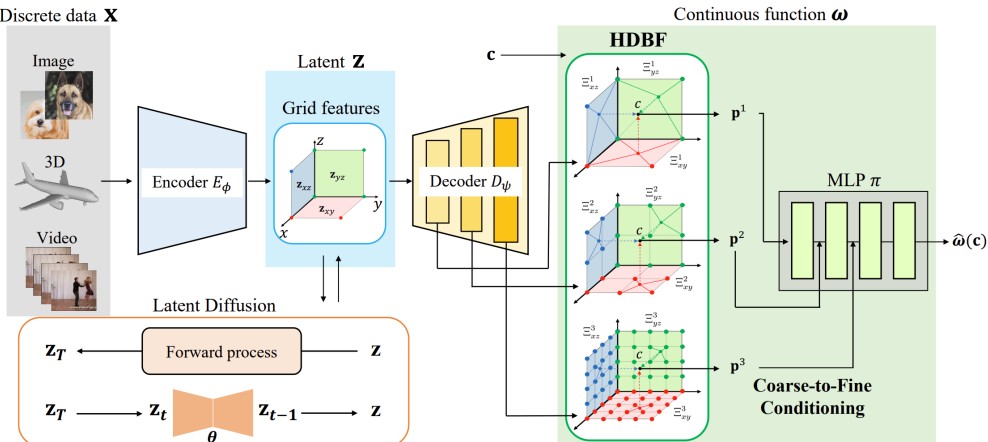

Figure 2: **Overall pipeline of DDMI**. Discrete data $\mathbf{x}$ and continuous function $\boldsymbol{\omega}$ are connected in the shared latent space $\mathbf{z}$ (D2C-VAE). The decoder generates Hierarchically-Decomposed Basis Fields (HDBFs) given latent variable $\mathbf{z}$. $\mathbf{p}^1$ represents the coarsest scale PE and $\mathbf{p}^3$ corresponds to the finest scale PE. The MLP returns the signal value for queried coordinate $c$ using the Coarse-to-Fine Conditioning method. Latent diffusion model operates on the shared latent space. Note that we use a tri-plane latent variable for 3D and video, and a single plane for 2D image.

known for their ability to generate data at arbitrary scales with a *single* model. Thus, there has been a surge of recent works across multiple modalities. For 2D images, CIPS (Anokhin et al., 2021) and INR-GAN (Skorokhodov et al., 2021) employ GANs to synthesize continuous image functions. For 3D shape, recent studies (Nam et al., 2022a; Zheng et al., 2022; Li et al., 2023; Erkoç et al., 2023) have proposed generating shapes as Signed Distance Functions (SDFs) using GANs or diffusion models. Also, for videos, DIGAN (Yu et al., 2022) and StyleGAN-V (Skorokhodov et al., 2022) have introduced GAN-based architectures to generate videos as continuous spatio-temporal functions. However, since these models are designed for specific modalities, they cannot easily adapt to different types of signals. Another line of research has explored the domain-agnostic architectures for INR generations, such as GASP (Dupont et al., 2022b), Functa (Dupont et al., 2022a), GEM (Du et al., 2021), and DPF (Zhuang et al., 2023). Zhuang et al. (2023) directly applies diffusion models to explicit signal fields to generate samples at the targeted modality, yet it faces a scalability issue when dealing with large-scale datasets. Others (Dupont et al., 2022b;a; Du et al., 2021; Koyuncu et al., 2023) attempt to model the weight distribution of INRs with GANs, diffusion models, or latent interpolation. In contrast, we introduce a domain-agnostic generative model that generates adaptive positional embeddings instead of weights of MLP in INRs.

**Latent diffusion model.** Diffusion models (Ho et al., 2020; 2022; Song et al., 2021) have demonstrated remarkable success in generation tasks. They consistently achieve high-quality results and high distribution coverage (Dhariwal & Nichol, 2021), often outperforming GANs. However, their iterative reverse process, typically involving a large number of steps (*e.g.*, 1000 steps), renders them significantly slower and inefficient compared to the implicit generative models like VAEs (Kingma & Welling, 2013) and GANs. To alleviate this limitation, recent works (Rombach et al., 2022; Vahdat et al., 2021) have proposed learning the data distribution in a low-dimensional latent space, which offers computational efficiency. Latent diffusion models (LDMs) strike a favorable balance between quality and efficiency, making them an attractive option for various applications. Our work adopts a latent space approach for designing the computational-efficient generative model.

## 3 METHODOLOGY

We present a Domain-agnostic latent Diffusion Model for synthesizing high-quality Implicit neural representations (DDMI). In order to learn to generate continuous functions (INRs) from discrete data, *e.g.*, images, we propose a novel VAE architecture D2C-VAE that maps discrete data to continuous functions via a shared latent space in Sec. 3.1. To enhance the quality of INR generation, we introduce a coarse-to-fine conditioning (CFC) mechanism for evaluating INRs with the generated hierarchically-decomposed basis fields (HDBFs). Sec. 3.2 outlines the two-stage training proce-

dures of the proposed method. As with existing latent diffusion models (Rombach et al., 2022), D2C-VAE learns the shared latent space in the first stage. In the second stage, the proposed framework trains a diffusion model in the shared latent space while keeping the other networks fixed. The overall pipeline is illustrated in Fig. 2.

### 3.1 DDMI

Let $\boldsymbol{\omega}, \hat{\boldsymbol{\omega}} \in \boldsymbol{\Omega}$ denote a continuous function representing an arbitrary signal and its approximation by neural networks respectively, where $\boldsymbol{\Omega}$ is a continuous function space. Given a spatial or spatiotemporal coordinate $c \in \mathbb{R}^m$, and its corresponding signal value $x \in \mathbb{R}^n$, training data $\mathbf{x}$ can be seen as the evaluations of a continuous function at a set of coordinates, i.e., $\mathbf{x} = [\boldsymbol{\omega}(c)]_{i=1}^I = \boldsymbol{\omega}(\mathbf{c})$, where $\mathbf{x} \in \mathbb{R}^{I \times n}$, $\mathbf{c} \in \mathbb{R}^{I \times m}$, and $I$ is the number of coordinates.

**D2C-VAE.** We propose an *asymmetric* VAE architecture, dubbed as Discrete-to-continuous space Variational Auto-Encoder (D2C-VAE), to seamlessly connect a discrete data space and a continuous function space via the shared latent space. Specifically, the encoder $E_\phi$ maps discrete data $\mathbf{x}$ to the latent variable $\mathbf{z}$ as 2D grid features, *e.g.*, a 2D plane for images or a 2D tri-plane for 3D shapes and videos. The decoder $D_\psi$ generates *basis fields* $\Xi$, *i.e.*, $\Xi = D_\psi(\mathbf{z})$, where we define $\Xi$ as a set of dense grids consisting of generated basis vectors. Then, the positional embedding $p$ for the coordinate $c$ is computed by $p = \gamma(c; \Xi)$, where a function $\gamma$ performs bilinear interpolation on $\Xi$, calculating the distance-weighted average of its four nearest basis vectors at coordinate $c$. For tri-plane basis fields, an axis-aligned orthogonal projection is applied beforehand (see Fig. 2). In this manner, the proposed method *adaptively* generates PEs according to different basis fields. Finally, the MLP $\pi_\theta$ returns the signal value given the positional embedding, *i.e.*, $\hat{x} = \pi_\theta(p) = \hat{\boldsymbol{\omega}}(c)$. Fig. 3 shows the distinction between the proposed method and existing INR generative models. We observed that the PE generation improves the expressive power of INRs compared to weight generation approaches, see Sec. 4.

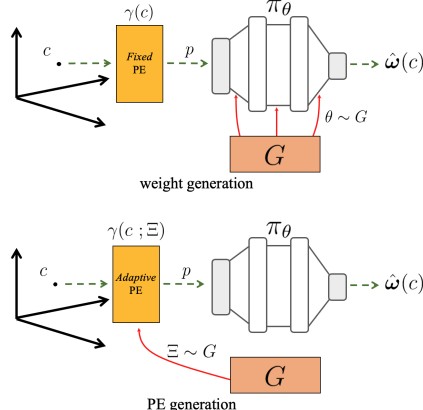

Figure 3: **Comparison between weight generation and PE generation for INR generative models** $G$. $c$ is a coordinate, p is a PE, $\gamma$ is a function that maps coordinates to PEs, $\pi_\theta$ is MLP, and $\hat{\boldsymbol{\omega}}(c)$ is a predicted signal value. For PE generation, we sample basis fields $\Xi$ from $G$ instead of $\theta$. The red line indicates the generation.

**Hierarchically-decomposed basis fields.** Instead of relying on a single-scale basis field, we propose decomposing it into multiple scales to better account for signals' multi-scale nature. To efficiently generate multi-scale basis fields, we leverage the feature hierarchy of a single neural network. Specifically, the decoder $D_\psi$ outputs feature maps at different scale $i$, *i.e.*, $D_\psi(\mathbf{z}) = \{\Xi^i | i = 1, ..., n\}$. The feature maps undergo $1 \times 1$ convolution to match the feature dimensions across scales. We refer to these decomposed basis fields as Hierarchically-Decomposed Basis Fields (HDBFs). Then, we can compute multi-scale PEs $\{p^i\}$ as $p^i = \gamma(c; \Xi^i)$, for all $i$. In practice, we use three different scales of basis fields from three different levels of layers. Sec. 4.4 qualitatively validates the spatial frequencies in HDBFs of generated samples are decomposed into multiple levels. This demonstrates that each field is dedicated to learning a specific level of detail, leading to a more expressive representation.

**Coarse-to-fine conditioning (CFC).** A naïve approach to using multi-scale positional embeddings from HDBFs for MLP $\pi_\theta$ is concatenating them along channel dimensions. However, we found that it is suboptimal. Thus, we introduce a conditioning mechanism that gradually conditions MLP $\pi_\theta$ on coarse PEs to fine PEs. The intuition is to encourage the lower-scale basis field to focus on the details missing from the higher-scale basis field. To achieve this, we feed the PE from the lowest-scale basis field as input to the MLP block and then concatenate (or element-wise sum) its intermediate output with the next PE from the higher-scale basis field. We continue this process for subsequent scales until reaching the $n$-th scale (see Fig. 2).

## 3.2 Training Procedure and Inference

The training of DDMI involves two stages: VAE training and diffusion model training. In the first stage, D2C-VAE learns the shared latent space with an encoder $E_\phi$ to map discrete data to latent vector $\mathbf{z}$ and a decoder $D_\psi$ to generate basis fields $\Xi$. In the second stage, a diffusion model is trained in the latent space to learn the empirical distribution of latent vectors $\mathbf{z}$.

**D2C-VAE training.** We define a training objective for D2C-VAE that maximizes the evidence lower bound (ELBO) of log-likelihood of the continuous function $\boldsymbol{\omega}$ with discrete data $\mathbf{x}$ as:

$$\log p(\boldsymbol{\omega}) = \log \int p_{\psi,\pi_\theta}(\boldsymbol{\omega}|\mathbf{z}) \cdot p(\mathbf{z})d\mathbf{z} \tag{1}$$

$$= \log \int \frac{p_{\psi,\pi_\theta}(\boldsymbol{\omega}|\mathbf{z})}{q_\phi(\mathbf{z}|\mathbf{x})} \cdot q_\phi(\mathbf{z}|\mathbf{x}) \cdot p(\mathbf{z}) \, d\mathbf{z} \tag{2}$$

$$\geq \int \log \left( \frac{p_{\psi,\pi_\theta}(\boldsymbol{\omega}|\mathbf{z})}{q_\phi(\mathbf{z}|\mathbf{x})} \cdot p(\mathbf{z}) \right) \cdot q_\phi(\mathbf{z}|\mathbf{x}) \, d\mathbf{z} \tag{3}$$

$$= \int q_\phi(\mathbf{z}|\mathbf{x}) \cdot \left( \log p_{\psi,\pi_\theta}(\boldsymbol{\omega}|\mathbf{z}) - \log \left( \frac{q_\phi(\mathbf{z}|\mathbf{x})}{p(\mathbf{z})} \right) \right) \tag{4}$$

$$= \mathbb{E}_{q_\phi(\mathbf{z}|\mathbf{x})} [\log p_{\psi,\pi_\theta}(\boldsymbol{\omega}|\mathbf{z})] - D_{KL}(q_\phi(\mathbf{z}|\mathbf{x})||p(\mathbf{z})), \tag{5}$$

where the inequality in Eq. 3 is by Jensen's inequality. $q_\phi(\mathbf{z}|\mathbf{x})$ is the approximate posterior, $p(\mathbf{z})$ is a prior, and $p_{\psi,\pi_\theta}(\mathbf{x}|\mathbf{z})$ is the likelihood. The first term in Eq. 5 measures the reconstruction loss, and the KL divergence between the posterior and prior distributions $p(\mathbf{z})$ encourages latent vectors to follow the prior. However, since we do not have observation $\boldsymbol{\omega}$ but only discrete data $\mathbf{x} = \boldsymbol{\omega}(\mathbf{c})$, we approximate $p_{\psi,\pi_\theta}(\boldsymbol{\omega}|\mathbf{z})$ by assuming coodinate-wise independence as

$$p_{\psi,\pi_\theta}(\boldsymbol{\omega}|\mathbf{z}) \approx p_{\psi,\pi_\theta}(\boldsymbol{\omega}(\mathbf{c})|\mathbf{z}) = \prod_{c\in\mathbf{c}} p_{\psi,\pi_\theta}(\boldsymbol{\omega}(c)|\mathbf{z}), \tag{6}$$

where $\hat{\boldsymbol{\omega}}(\mathbf{c}) = \pi_\theta(\gamma(\mathbf{c}, D_\psi(\mathbf{z})))$. Thus, our training objective in Eq. 5 can be approximated as

$$L_{\phi,\psi,\theta}(\mathbf{x}) := \mathbb{E}_{q_\phi(\mathbf{z}|\mathbf{x})} [\log p_{\psi,\pi_\theta}(\boldsymbol{\omega}|\mathbf{z})] - D_{KL}(q_\phi(\mathbf{z}|\mathbf{x})||p(\mathbf{z})) \tag{7}$$

$$\approx \mathbb{E}_{q_\phi(\mathbf{z}|\mathbf{x})} \left[ \sum_{c\in\mathbf{c}} \log p_{\psi,\pi_\theta}(\boldsymbol{\omega}(c)|\mathbf{z}) \right] - D_{KL}(q_\phi(\mathbf{z}|\mathbf{x})||p(\mathbf{z})). \tag{8}$$

Since the reconstruction loss varies depending on the number of coordinates, *i.e.*, $|\mathbf{c}|$, in practice, we train D2C-VAE with the re-weighted objective function given as:

$$L_{\phi,\psi,\pi_\theta}(\mathbf{x}) = \mathbb{E}_{q_\phi(\mathbf{z}|\mathbf{x}),c\in\mathbf{c}} [\log p_{\psi,\pi_\theta}(\boldsymbol{\omega}(c)|\mathbf{z})] - \lambda_z \cdot D_{KL}(q_\phi(\mathbf{z}|\mathbf{x})||p(\mathbf{z})), \tag{9}$$

where $\lambda_{\mathbf{z}}$ balances the two losses, and $p(\mathbf{z})$ is a standard normal distribution.

**Diffusion model training.** Following existing LDMs (Vahdat et al., 2021; Rombach et al., 2022; Ho et al., 2020), the forward diffusion process is defined in the learned latent space as a Markov chain with pre-defined Gaussian kernels $q(\mathbf{z}_t|\mathbf{z}_{t-1}) := \mathcal{N}(\mathbf{z}_t; \sqrt{1-\beta_t}\mathbf{z}_{t-1}, \beta_t\mathbf{I})$. The forward diffusion process is given as $q(\mathbf{z}_{1:T}|\mathbf{z}_0) = \prod_{t=1}^{T} q(\mathbf{z}_t|\mathbf{z}_{t-1})$, where $T$ indicates the total number of diffusion steps and $\beta_t$ is a pre-defined noise schedule that satisfies $q(\mathbf{z}_T) \approx \mathcal{N}(\mathbf{z}_T; 0, \mathrm{I})$. The reverse diffusion process is also defined in the latent space as $p_\varphi(\mathbf{z}_{0:T}) = p(\mathbf{z}_T) \prod_{t=1}^{T} p_\varphi(\mathbf{z}_{t-1}|\mathbf{z}_t)$, where $p_\varphi(\mathbf{z}_0)$ is a LDM prior. The Gaussian kernels $p_\varphi(\mathbf{z}_{t-1}|\mathbf{z}_t) := \mathcal{N}(\mathbf{z}_t; \mu_\varphi(\mathbf{z}_t, t), \rho_t^2\mathrm{I})$ is parameterized by a neural network $\mu_\varphi(\mathbf{z}_t, t)$ and $\rho_t^2$ is the fixed variances. The reverse diffusion process $p_\varphi(\mathbf{z}_{t-1}|\mathbf{z}_t)$ is trained with the following noise prediction using reparameterization trick:

$$L_\varphi(\mathbf{z}) = \mathbb{E}_{\mathbf{z}_0,\epsilon,t} \left[ w(t)||\epsilon - \epsilon_\varphi(\mathbf{z}_t, t)||_2^2 \right]. \tag{10}$$

**Inference.** Continuous function generation involves the reverse diffusion process $p_\varphi$, D2C-VAE decoder $D_\psi$, and read-out MLP $\pi_\theta$. First, a latent $\mathbf{z}_0 \sim p_\varphi(\mathbf{z}_0)$ is generated by iteratively conducting ancestral sampling, $\mathbf{z}_{t-1} = \frac{1}{\sqrt{\alpha_t}} \left( \mathbf{z}_t - \frac{\beta_t}{\sqrt{1-\bar{\alpha}_t}}\epsilon_\varphi(\mathbf{z}_t, t) \right) + \sigma_t\zeta$, where $\zeta \sim \mathcal{N}(0, \mathrm{I})$ and $p(\mathbf{z}_T) = \mathcal{N}(\mathbf{z}_T; 0, \mathrm{I})$. Then, the latent $\mathbf{z}_0$ is fed to D2C-VAE decoder $D_\psi$ to generate HDBFs $\Xi$. Finally, combining HDBFs with MLP $\pi_\theta$ learned in the first stage, the proposed method parameterizes a continuous function $\boldsymbol{\omega}(\cdot)$.

Table 1: FID results on CelebA-HQ.

| | $64^2$ | $128^2$ | $256^2$ | $384^2$ |
|---|---|---|---|---|
| <Discrete representation> | | | | |
| LDM (Rombach et al., 2022) | - | - | 5.51 | - |
| LSGM (Vahdat et al., 2021) | - | - | 7.22 | - |
| <Continuous representation> | | | | |
| Domain-specific | | | | |
| INR-GAN (Skorokhodov et al., 2021) | - | - | 10.3 | - |
| CIPS (Anokhin et al., 2021) | 15.41 | 13.53 | 11.4 | 15.8 |
| Domain-agnostic | | | | |
| Functa (Dupont et al., 2022a) | 40.4 | - | - | - |
| GEM (Du et al., 2021) | 30.4 | - | - | - |
| GASP (Dupont et al., 2022b) | 13.5 | 19.2 | - | - |
| DPF (Zhuang et al., 2023) | 13.2 | - | - | - |
| **DDMI (Ours)** | **9.74** | **8.73** | **7.25** | **10.44** |

Table 3: FID results on AFHQv2 Dog.

| | $128^2$ | $256^2$ | $384^2$ |
|---|---|---|---|
| <Discrete representation> | | | |
| StyleGAN (Karras et al., 2020a) | - | 6.73 | - |
| <Continuous representation> | | | |
| Domain-specific | | | |
| INR-GAN (Skorokhodov et al., 2021) | - | 31.27 | - |
| CIPS (Anokhin et al., 2021) | 26.95 | 23.93 | 28.97 |
| Domain-agnostic | | | |
| GASP (Dupont et al., 2022b) | - | 35.78 | - |
| **DDMI (Ours)** | **10.81** | **8.54** | **11.47** |

Table 4: FID results on AFHQv2 Cat.

| | $128^2$ | $256^2$ | $384^2$ |
|---|---|---|---|
| <Discrete representation> | | | |
| StyleGAN (Karras et al., 2020a) | - | 3.25 | - |
| <Continuous representation> | | | |
| Domain-specific | | | |
| INR-GAN (Skorokhodov et al., 2021) | - | 11.2 | - |
| CIPS (Anokhin et al., 2021) | 7.85 | 7.35 | 11.8 |
| Domain-agnostic | | | |
| GASP (Dupont et al., 2022b) | - | 17.48 | - |
| **DDMI (Ours)** | **5.88** | **4.27** | **7.94** |

Table 2: Precision and Recall results.

| | CelebA-HQ | | AFHQv2 Cat | |
|---|---|---|---|---|
| <Continuous representation> | P↑ | R↑ | P↑ | R↑ |
| INR-GAN (Skorokhodov et al., 2021) | 0.671 | 0.333 | 0.719 | 0.281 |
| CIPS (Anokhin et al., 2021) | 0.682 | 0.287 | 0.716 | 0.117 |
| **DDMI (Ours)** | **0.734** | **0.408** | **0.808** | **0.367** |

## 4 EXPERIMENTS

We evaluate the effectiveness and versatility of DDMI through comprehensive experiments across diverse modalities, including 2D images, 3D shapes, and videos. We assume a multivariate normal distribution for the likelihood function $q_\phi(\boldsymbol{\omega}(c)|\mathbf{z})$ for images and videos and Bernoulli distribution for shapes (occupancy function). For all domains, the multivariate normal distribution is used for the posterior $p_{\psi,\pi_\theta}(\mathbf{z}|\mathbf{x})$. For the type of latent variables for each domain, the encoder maps input data to the latent variable $\mathbf{z}$ as 2D grid features, *e.g.*, a single 2D plane for images using a 2D CNN-based encoder (Ho et al., 2020) or a 2D tri-plane for 3D shapes following Conv-ONET (Peng et al., 2020) and videos as outlined in Timesformer (Bertasius et al., 2021). Then, we use a 2D CNN-based decoder $D_\psi$ to convert latent variable $\mathbf{z}$ into basis fields $\Xi$. Additional experiments (*e.g.*, NeRF) and more implementation details, evaluation, and baselines are provided in the supplement.

### 4.1 2D IMAGES

**Datasets and baselines.** For images, we evaluate models on AFHQv2 Cat and Dog (Choi et al., 2020) and CelebA-HQ dataset (Karras et al., 2018) with a resolution of $256^2$. We compared our method with three groups of models: 1) *Domain-agnostic* INR generative models such as Functa (Dupont et al., 2022a), GEM (Du et al., 2021), GASP (Dupont et al., 2022b), and DPF (Zhuang et al., 2023), which are the *primary* baselines, 2) *Domain-specific* INR generative models that are specifically tailored for image generation like INR-GAN (Skorokhodov et al., 2021) and CIPS (Anokhin et al., 2021). Apart from DPF, which generates the explicit signal field, every baseline operates weight generation, whereas ours opts for PE generation. 3) *Discrete representation* based generative models for reference. We provide results from state-of-the-art generative models (Vahdat et al., 2021; Rombach et al., 2022; Karras et al., 2020a) that learn to generate discrete images.

**Quantitative analysis.** We primarily measure FID (Heusel et al., 2017), following the setup in (Skorokhodov et al., 2021). Tab. 1, 3, and 4 showcase the consistent improvement of DDMI over primary baselines for multiple resolutions. For instance, Tab. 1 shows that compared to DPF, the recent domain-agnostic INR generative model, our approach achieves an FID score of 9.74 as opposed to 13.2 on the CelebA-HQ at a resolution of $64^2$. Moreover, on AFHQv2 Cat in Tab. 4, DDMI demonstrates superior performance over CIPS, specifically developed for arbitrary scale image generation, achieving an average FID improvement of 2.97 across three different resolutions. In Tab. 2, we compare precision and recall with image-targeted INR generative models. Here, precision and recall are indicative of fidelity and diversity, respectively. DDMI exhibits a significant advantage over both baselines (Skorokhodov et al., 2021; Anokhin et al., 2021), demonstrating superior performance for both CelebA-HQ and AFHQv2 Cat datasets. We provide additional results on CIFAR10 (Krizhevsky et al., 2009) and Lsun Churches (Yu et al., 2015) in Tab. 11 of the supplement.

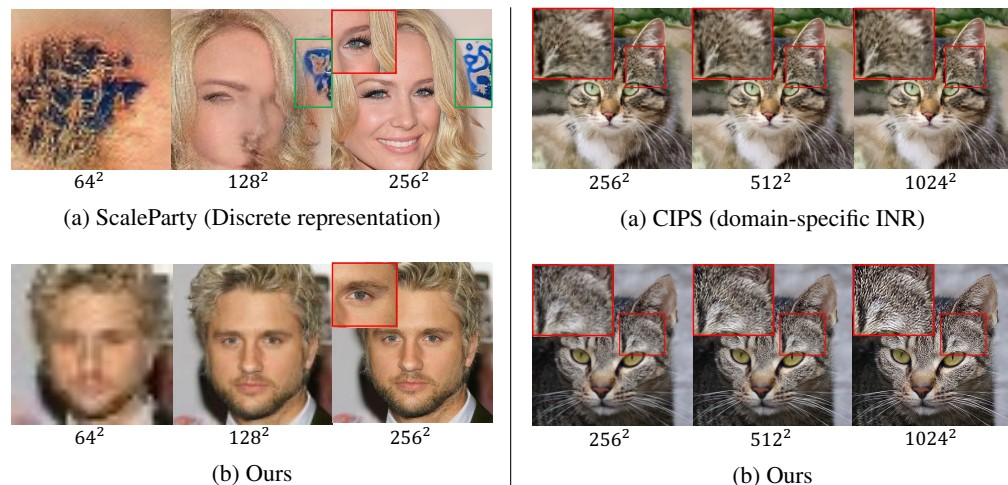

<table>
<tr><td>

(a) ScaleParty (Discrete representation)

(b) Ours

Figure 4: Comparison between DDMI and ScaleParty on arbitrary-scale generation.

</td><td>

(a) CIPS (domain-specific INR)

(b) Ours

Figure 5: Comparison between DDMI and CIPS on arbitrary-scale generation.

</td></tr>
</table>

**Qualitative analysis.** For further validation, we generate images at arbitrary scales and conduct comparisons with two notable models: ScaleParty (Ntavelis et al., 2022), a recent generative model designed for multi-resolution discrete images, and CIPS, an image-targeted INR generative model. DDMI demonstrates an impressive capability to consistently generate clearer images across various resolutions, from low to high. In Fig. 4, our model excels at generating images with preserved facial structure, whereas Ntavelis et al. (2022) struggles to maintain the global structure of the image for lower-resolution cases. Also, in Fig. 5, DDMI succeeds in capturing high-frequency details across images of varying resolutions, while Anokhin et al. (2021) tends to lack finer details as the resolution increases.

## 4.2 3D SHAPES

**Datasets and baselines.** For shapes, we adopt the ShapeNet dataset (Chang et al., 2015) with two settings: a single-class dataset with 4K chair shapes and a multi-class dataset comprising 13 classes with 35K shapes, following the experimental setup in (Peng et al., 2020). We learn the shape as the occupancy function (Mescheder et al., 2019) $\omega : \mathbb{R}^3 \rightarrow \{0, 1\}$, where it maps 3D coordinates to occupancy values, *e.g.*, 0 or 1. Still, domain-agnostic INR generative models from Sec. 4.1 are our primary baselines, where we also compare with domain-specific INR generative models for 3D shapes like 3D-LDM (Nam et al., 2022a) and SDF-Diffusion (Shim et al., 2023). We also include results from generative models for generating discrete shapes, such as point clouds.

**Quantitative analysis.** We conduct extensive experiments with unconditional and conditional shape generation. Beginning with unconditional shape generation (Tab. 5), we measure fidelity and diversity using Minimum Matching Distance (MMD) and Coverage (Achlioptas et al., 2018). DDMI surpasses both domain-agnostic and domain-specific baselines, achieving the best MMD for the single chair class (1.5) and multi-class (1.3) settings. Moreover, we attain the highest COV in multi-class shape generation, showcasing our ability to generate diverse shapes with high fidelity.

Next, we provide text-guided shape generation results on Text2Shape (T2S) dataset (Chen et al., 2019), comprising 75K paired examples of text and shapes on chairs and tables. For training, we utilize pre-trained CLIP text encoder (Radford et al., 2021) $\tau$ for encoding text prompt $t$ into embedding $\tau(t)$ and condition it to LDM $e_\theta(z_t, t, \tau(t))$ by cross-attention (Rombach et al., 2022). For generation, the text-conditioned score $\hat{e}$ is derived using classifier-free guidance (Ho & Salimans, 2022): $\hat{e}_\theta(z_t, t, \tau(t)) = e_\theta(z_t, t, \emptyset) + w \cdot (e_\theta(z_t, t, \tau(t)) - e_\theta(z_t, t, \emptyset))$, where $\emptyset$ and $w$ indicates an empty prompt and guidance scale, respectively. We measure classification accuracy, CLIP-S (CILP similarity score), and Total Mutual Difference (TMD) to evaluate against shape generative models (Mittal et al., 2022; Li et al., 2023; Liu et al., 2022). Tab. 7 illustrates the strong performance of

Table 5: Generation results on 3D shapes.

| | Chair | | Multi Class | |
|---|---|---|---|---|
| | MMD ↓ | COV ↑ | MMD ↓ | COV ↑ |
| <Discrete representation> | | | | |
| PVD (Zhou et al., 2021) | 6.8 | 0.421 | - | - |
| DPM3D (Luo & Hu, 2021) | 1.3* | 0.567* | - | - |
| | | | | |
| <Continuous representation> | | | | |
| Domain-specific | | | | |
| LatentGAN (Chen & Zhang, 2019) | - | - | 1.7 | 0.389 |
| 3D-LDM (Nam et al., 2022a) | 1.68 | 0.426 | - | - |
| SDF-StyleGAN (Zheng et al., 2022) | 1.9 | 0.411 | 1.55 | 0.398 |
| SDF-Diffusion (Shim et al., 2023) | 8.0 | 0.498 | - | - |
| HyperDiffusion (Erkoç et al., 2023) | 7.1 | **0.530** | - | - |
| Domain-agnostic | | | | |
| GASP (Dupont et al., 2022b) | 2.5 | 0.353 | 2.1 | 0.341 |
| GEM (Du et al., 2021) | - | - | 1.4 | 0.409 |
| DPF (Zhuang et al., 2023) | - | - | 1.6 | 0.419 |
| **DDMI (Ours)** | **1.5** | 0.510 | **1.3** | **0.421** |

∗ are trained on Acronym (Eppner et al., 2021) dataset.

Table 6: Generation results on videos.

| | SkyTimelapse |
|---|---|
| | FVD ↓ |
| <Discrete representation> | |
| VideoGPT (Yan et al., 2021) | 222.7 |
| MoCoGAN (Tulyakov et al., 2018) | 206.6 |
| MoCoGAN-HD (Tian et al., 2021) | 164.1 |
| LVDM (He et al., 2022) | 95.2 |
| PVDM (Yu et al., 2023) | 71.46 |
| | |
| <Continuous representation> | |
| Domain-specific | |
| DIGAN (Yu et al., 2022) | 83.11 |
| StyleGAN-V (Skorokhodov et al., 2022) | 79.52 |
| Domain-agnostic | |
| **DDMI (Ours)** | **66.25** |

Table 7: Text-to-shape generation results.

| | Acc ↑ | CLIP-S ↑ | TMD ↓ |
|---|---|---|---|
| Domain-specific | | | |
| IMLE (Liu et al., 2022) | 34.79 | - | 0.891 |
| Auto-SDF (Mittal et al., 2022) | 83.88 | - | 0.581 |
| Diffusion-SDF (Li et al., 2023) | 88.56 | 28.63 | **0.169** |
| Domain-agnostic | | | |
| **DDMI (Ours, $w = 3$)** | **91.30** | **30.30** | 0.204 |

Table 8: Ablation study.

| Generation target | HDBFs | CFC | First Stage (PSNR) | Second Stage (FID) |
|---|---|---|---|---|
| PE | | | 32.72 | 8.54 |
| PE | ✓ | | 33.17 | 8.23 |
| PE | ✓ | ✓ | **33.56** | **7.82** |

DDMI, indicating our method generates high-fidelity shapes (accuracy and CLIP-S) with a diversity level comparable to baselines (TMD).

**Qualitative analysis.** Fig. 1 and Fig.6 present qualitative results. Fig. 6 displays visualizations of text-guided shape generations, demonstrating our model's consistent generation of faithful shapes given text conditions (*e.g.*, "a two-layered table") over Diffusion-SDF (Li et al., 2023). Fig. 1 incorporates the comprehensive results, including unconditional shape generation, text-conditioned shape generation, and unconditional Neural Radiance Field (NeRF) generation. Especially for NeRF, we train DDMI with SRN Cars dataset (Sitzmann et al., 2019). Specifically, we encode point clouds to the 2D-triplane HDBFs, allowing MLP to read out triplane features at queried coordinate and ray direction into color and density via neural rendering (Mildenhall et al., 2021). For more results, including a qualitative comparison between Functa (Dupont et al., 2022a) and ours on NeRF generation, see Fig. 15 in the supplement.

## 4.3 VIDEOS

**Datasets and baselines.** For videos, we use the SkyTimelapse dataset (Xiong et al., 2018) and preprocess each video to have 16 frames and a resolution of $256^2$, following the conventional setup used in recent video generative models (Skorokhodov et al., 2022; Yu et al., 2023). We learn 2D videos as a continuous function $\omega$ maps spatial-temporal coordinates to corresponding RGB values, *i.e.*, $\omega : \mathbb{R}^3 \to \mathbb{R}^3$. We compare our results with domain-specific INR generative models (Yu et al., 2022; Skorokhodov et al., 2022) as well as discrete video generative models like LVDM (He et al., 2022) and PVDM (Yu et al., 2023).

**Quantitative analysis.** Tab. 6 illustrates the quantitative results of 2D video generation. We employ Fréchet Video Distance (FVD) (Unterthiner et al., 2018) as the evaluation metric, following StyleGAN-V (Skorokhodov et al., 2022). DDMI shows competitive performance to the most recent diffusion model (PVDM (Yu et al., 2023)), which is specifically trained on discrete video (pixels). This validates the effectiveness of our design choices in enhancing the expressive power of INR. We also provide qualitative results in Fig. 1 and 14, where the generated videos exhibit realistic quality in each frame and effectively capture the motion across frames, demonstrating the versatility of our model in handling not only spatial but also temporal dimensions.

## 4.4 ANALYSIS

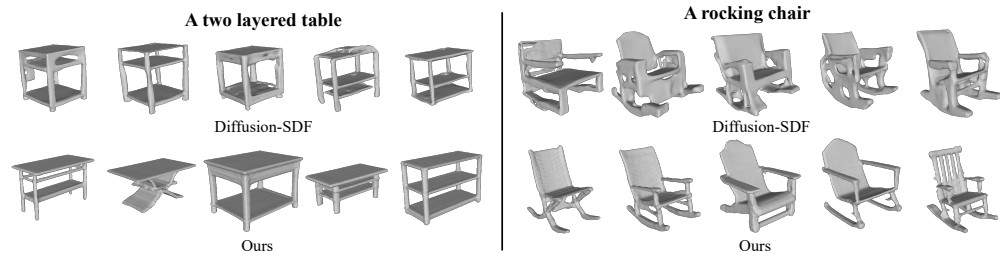

Figure 6: **Qualitative comparison on text-guided shape generation.** We present a comparison of the generation results produced by Diffusion-SDF and DDMI for given text prompt. DDMI excels in generating intricate details while preserving smooth surfaces. In contrast, Diffusion-SDF struggles to capture fine details and often produces less polished surfaces.

**Decomposition of HDBFs.** In Fig. 7, we analyze the role of different scales of HDBFs in representing signals. Specifically, we zero out all HDBFs except one during generation and observe the results in both spatial and spectral domains. When (a) $\Xi^2$ and $\Xi^3$ are zeroed out, the generated image contains coarser details, such as colors, indicating that the first $\Xi^1$ focuses on larger-scale components. In contrast, (c) with the third $\Xi^3$, our model generates images of high-frequency details, such as whiskers and furs. Results in the spectral domain after Fourier transform also match the tendency in the spatial domain, as the spectra of (c) exhibit high magnitudes in the high-frequency domains, whereas those of (a) have high magnitudes in the low-frequency domains. The analysis indicates that HDBFs effectively decompose basis fields to capture the signals of different scales.

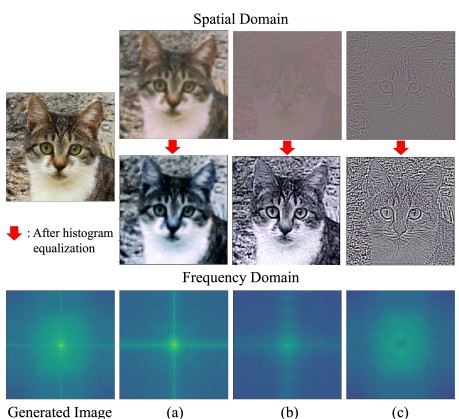

Figure 7: **HDBFs analysis.** The upper two rows show generated images whereas the bottom one shows the spectral magnitude after applying Fourier transform. The far left column indicates the generated image. Other columns indicate the generated image with (a): $\Xi^2, \Xi^3$, (b): $\Xi^1, \Xi^3$, and (c): $\Xi^1, \Xi^2$ zeroed out, respectively. We employ histogram equalization for better visualization.

**Ablation.** Here, we conduct an ablation study to evaluate the impact of each component in DDMI. We train DDMI using various configurations on the AFHQv2 Cat dataset with a $384^2$ resolution. Refer to Tab. 8 for the specific configurations. The baseline refers to D2C-VAE without HDBFs and CFC. We incrementally introduce components and present the Peak Signal-to-Noise Ratio (PSNR) and FID scores at the end of the first and second training stages. The results, outlined in Tab. 8, reveal a gradual increase in PSNR and a corresponding decrease in FID scores as we incorporate additional components. This underscores the effectiveness of these components in enhancing the fidelity and realism of INR.

## 5 CONCLUSION

In this paper, we propose DDMI, a domain-agnostic latent diffusion model designed to synthesize high-quality Implicit Neural Representations (INRs) across various signal domains. Our approach defines the Discrete-to-continuous space Variational AutoEncoder (D2C-VAE), which generates Positional Embeddings (PEs) and establishes a seamless connection between discrete data and continuous functions. Leveraging this foundation and our novel coarse-to-fine conditioning mechanism with Hierarchically Decomposed Basis Fields (HDBFs), our extensive experiments across a wide range of domains have consistently demonstrated the versatility and superior performance of DDMI when compared to existing INR-based generative models.

ACKNOWLEDGEMENT

This work was supported in part by ICT Creative Consilience program (IITP2024-2020-0-01819) supervised by the IITP, the National Research Foundation of Korea (NRF) grant funded by the Korea government (MSIT) (NRF-2023R1A2C2005373), and the Virtual Engineering Platform Project (Grant No. P0022336), funded by the Ministry of Trade, Industry & Energy (MoTIE, South Korea).

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

## A    IMPLEMENTATION DETAILS

We here provide more details of our implementation, including the architecture of our models and hyperparameters. The comprehensive lists of hyperparameters used in experiments are provided in Tab. 9.

**Encoders and latent spaces.**    We use slightly different encoder backbones and structures for the latent variable depending on the signal domains. For a 2D image $\mathbf{x}$, we encode it into a latent variable $\mathbf{z}$ as a 2D plane, using a 2D CNN-based U-Net encoder (Ho et al., 2020).

For a 3D point cloud input $\mathbf{x} \in \mathbb{R}^{3 \times N}$, consisting of $N$ points in $xyz$-coordinates, we encode it into a tri-plane latent variable $\mathbf{z} = \{\mathbf{z}_{xy}, \mathbf{z}_{yz}, \mathbf{z}_{xz}\}$, following Conv-ONet (Peng et al., 2020). Specifically, given an input point cloud, we employ PointNet (Qi et al., 2017) for feature extraction and apply an orthographic projection to map its feature onto three canonical planes. Features projecting onto the same pixel are aggregated using a local pooling operation, resulting in three distinct 2D grid features. We then apply a shared 2D CNN encoder to the projected features on grid features to obtain the tri-plane latent variables.

Lastly, for 2D video $\mathbf{x}$, we encode it into a latent variable as a 2D tri-plane. We first convert it into 3D features using TimesFormer (Bertasius et al., 2021) encoder. Then, we perform 2D projection onto three canonical planes for latent variables $\mathbf{z} = [\mathbf{z}_{xy}, \mathbf{z}_{ys}, \mathbf{z}_{xs}]$ using three separate small Transformers (Vaswani et al., 2017), respectively, following Yu et al. (2023). Note that $s$ denotes a temporal dimension.

**Decoders.**    For all domains, we use 2D CNN-based U-Net decoder $D_\psi$ to convert latent variable $\mathbf{z}$ into basis fields $\Xi$. We maintain the structure of the basis fields to be consistent with the structure of the latent variables. For instance, in the case of 3D shapes, we compute tri-plane basis fields as $\Xi = \{\Xi_{xy}, \Xi_{yz}, \Xi_{xz}\}$, where $\Xi_{xy} = D_\psi(\mathbf{z}_{xy}), \Xi_{yz} = D_\psi(\mathbf{z}_{yz})$, and $\Xi_{xz} = D_\psi(\mathbf{z}_{xz})$. The same procedure is applied to other data domains, *e.g.*, $\Xi = \Xi_{xy}$ for images and $\Xi = [\Xi_{xy}, \Xi_{ys}, \Xi_{xs}]$ for videos. As mentioned in the main paper, we use three different scales of basis fields ($\Xi^1$, $\Xi^2$, and $\Xi^3$) from our HDBFs.

**MLP function.**    Given the basis fields $\Xi$ computed above, we compute positional embedding $p$ for given coordinate $c$ by bilinear interpolation that uses the distance-weighted average of the four nearest features of $\Xi$ from the coordinate $c$. We apply the same procedure to each scale of basis fields for a given coordinate to achieve $p^1$, $p^2$, and $p^3$. For tri-plane $\Xi$, an axis-aligned orthogonal projection for each plane is applied beforehand. Then the positional embedding $p$ at $c$ is fed into MLP-based residual blocks $n_\pi$ with $w_\pi$ channels.

**Latent diffusion model.**    For the LDM backbone, we adopt the 2D Unet model LDM (Rombach et al., 2022) and modify the hyperparameters.

## B    TRAINING AND INFERENCE

Our framework performs two-stage training. In the first stage, the proposed method learns the latent space by optimizing D2C-VAE. In the second stage, we optimize a diffusion model in the learned latent space. In this work, all experiments are conducted on 8 NVIDIA RTX3090 and 8 V100 GPUs. We provide the hyperparameters of training in Tab. 10.

### B.1    FIRST-STAGE TRAINING.

In the first stage, we optimize the encoder, decoder, and MLP of D2C-VAE using the AdamW (Loshchilov & Hutter, 2017) optimizer. We minimize the re-weighted ELBO objective of D2C-VAE in Eq. 9. We assume the standard normal prior distribution $p(\mathbf{z})$ and a multivariate normal distribution for the posterior $q_\phi(\mathbf{z}|\mathbf{x})$. The KL divergence loss between the posterior and the prior is calculated using the reparameterization trick Kingma & Welling (2013). *For image, video, and NeRF*, we assume that $p_{\psi, \pi_\theta}(\boldsymbol{\omega}(c)|\mathbf{z})$ follows the multivariate normal distribution with isotropic

Table 9: Hyperparameters of models.

| | AFHQ Cat/Dog $256^2$ | CelebA-HQ $256^2$ | CIFAR10 $32^2$ | Lsun Church $128^2$ | ShapeNet Chair | ShapeNet Multi Class | SKY $256^2$ |
|---|---|---|---|---|---|---|---|
| **Encoder** | | | | | | | |
| latent $\mathbf{z}$ shape | 2D plane | 2D plane | 2D plane | 2D plane | 2D tri-plane | 2D tri-plane | 2D tri-plane |
| dims. of $\mathbf{z}$ | $64^3$ | $64^3$ | $16^2 \cdot 64$ | $32^2 \cdot 64$ | $16^2 \cdot 64$ | $16^2 \cdot 64$ | $64^3$ |
| **Decoder** | | | | | | | |
| channel multiplier | (4,2,1) | (4,2,1) | (4,2,1) | (4,2,1) | (4,2,1) | (4,2,1) | (8,4,2,1) |
| head channel | 64 | 64 | 64 | 64 | 32 | 32 | 64 |
| basis field $\Xi$ shape | 2D plane | 2D plane | 2D plane | 2D plane | 2D tri-plane | 2D tri-plane | 2D tri-plane |
| basis field res. | (1,2,4) | (1,2,4) | (1) | (1,2,4) | (1,2,4) | (1,2,4) | (1,2,4) |
| **MLP** | | | | | | | |
| $n_\pi$ | 4 | 4 | 4 | 4 | 4 | 4 | 4 |
| $w_\pi$ | 256 | 256 | 256 | 256 | 256 | 256 | 256 |
| **Latent diffusion model** | | | | | | | |
| residual blocks | 2 | 2 | 2 | 2 | 2 | 2 | 2 |
| channel multiplier | (4,3,2,1) | (4,3,2,1) | (4,3,2,1) | (4,3,2,1) | (4,3,2,1) | (4,3,2,1) | (4,3,2,1) |
| head channel | 256 | 256 | 128 | 256 | 256 | 256 | 256 |
| attn res. | (2,4,8) | (2,4,8) | (2,4) | (2,4,8) | (2,4,8) | (2,4,8) | (2,4,8) |

Table 10: Training details

| | AFHQ Cat/Dog $256^2$ | CelebA-HQ $256^2$ | CIFAR10 $32^2$ | Lsun Church $128^2$ | ShapeNet Chair | ShapeNet Multi Class | SKY $256^2$ |
|---|---|---|---|---|---|---|---|
| **First stage training** | | | | | | | |
| learning rate | 1e-4 | 1e-4 | 1e-4 | 1e-4 | 1e-4 | 1e-4 | 1e-4 |
| optimizer | AdamW | AdamW | AdamW | AdamW | AdamW | AdamW | AdamW |
| SN weight start | 10 | 10 | 10 | 10 | 10 | 10 | 10 |
| SN weight decay anneal | linear | linear | linear | linear | linear | linear | linear |
| $\lambda_\mathbf{z}$ start | 1e-4 | 1e-4 | 1e-4 | 1e-4 | 1e-4 | 1e-4 | 1e-4 |
| $\lambda_\mathbf{z}$ end | 1e-3 | 1e-3 | 1e-3 | 1e-3 | 1e-3 | 1e-3 | 1e-3 |
| $\lambda_\mathbf{z}$ anneal | linear | linear | linear | linear | linear | linear | linear |
| $\lambda_\mathbf{z}$ anneal portion | 0.9 | 0.9 | 0.9 | 0.9 | 0.9 | 0.9 | 0.9 |
| **Second stage training** | | | | | | | |
| learning rate | 1e-4 | 1e-4 | 1e-4 | 1e-4 | 2e-4 | 2e-4 | 1e-4 |
| optimizer | AdamW | AdamW | AdamW | AdamW | AdamW | AdamW | AdamW |

covariance $\sigma$:

$$p_{\psi,\pi_\theta}(\boldsymbol{\omega}(\mathbf{c})|\mathbf{z}) = \prod_{c \in \mathbf{c}} \frac{1}{Z} \exp\left(-\frac{||\boldsymbol{\omega}(c) - \hat{\boldsymbol{\omega}}(c)||_2^2}{2\sigma^2}\right), \ \boldsymbol{\omega}(c) \in \mathbb{R}. \tag{11}$$

For the *occupancy function*, we assume the Bernoulli distribution:

$$p_{\psi,\pi_\theta}(\boldsymbol{\omega}(\mathbf{c})|\mathbf{z}) = \prod_{c \in \mathbf{c}} \hat{\boldsymbol{\omega}}(c)^{\boldsymbol{\omega}(c)} \cdot (1 - \hat{\boldsymbol{\omega}}(c))^{\boldsymbol{\omega}(c)}, \ \boldsymbol{\omega}(c) \in \{0,1\}. \tag{12}$$

In practice, we employ $\ell_1$-loss for image, video, and NeRF, and binary cross-entropy for 3D shapes. We linearly increase the weight $\lambda_\mathbf{z}$ during training and apply a spectral normalization (SN) regularization for the encoder, following Vahdat et al. (2021).

**Multi-scale training and scale injection.** For training our model on images, we incorporate the multi-scale training scheme (Ntavelis et al., 2022) to further encourage the estimation of accurate $\hat{\boldsymbol{\omega}}$. In our approach, we incorporate a scale variable $s \in \mathbb{R}$ that represents the sampling period of the coordinate set $\mathbf{c}$ for the discrete data $\mathbf{x}$. Through the training of D2C-VAE with data of diverse scales, *e.g.*, utilizing multi-resolution data or applying random crop augmentation, we enable the INR to explore a wider range of coordinate sets $\mathbf{c}$ in Eq. 9, enabling high-quality generation at arbitrary scales. In addition to multi-scale training, we incorporate a Scale Injection (SI) to enhance the performance of our framework further. We modulate the weights of each fully connected layer of MLP with the scale variable $s$, similar to style modulation in StyleGAN Karras et al. (2020b). The injection of scale information makes INR aware of the target scale. Specifically, the $(i, j)$ entry of $l_{th}$ layer weight $\phi^l \in \mathbb{R}^{m \times n}$ is modulated to $\hat{\phi}^l$:

$$\hat{\phi}_{i,j}^l(s) = \frac{\phi_{i,j}^l \cdot A^l(s)_j}{\sqrt{\Sigma_k \left(\phi_{i,k}^l \cdot A^l(s)_k\right)^2 + \varepsilon}}, \tag{13}$$

where $\varepsilon$ is a small constant for numerical stability and $A^l$ indicates layer-wise mapping function. $s$ undergoes Fourier embedding with a fully connected layer before the layer-wise mapping. In practice, we provide the encoder with full-frame fixed-resolution images of size $r \times r$ (*e.g.*, $256 \times 256$) to make fixed-size latent variables. Then, for each batch of latent variables, we randomly select a resolution from a predefined set of resolutions $\{r, 1.5 \times r, 2 \times r\}$ that the INR aims to present in a discrete image. For images with a higher resolution than $r$, we randomly crop them to match the resolution $r$.

## B.2 SECOND-STAGE TRAINING.

In the second stage, we optimize LDM on the empirical distribution of the latent variables using the AdamW (Loshchilov & Hutter, 2017) optimizer. We minimize the noise prediction loss, discussed in the main paper, while the other networks trained in the first stage are frozen. *In the case of triplane latent variables*, *e.g.*, 3D and video, we use a single 2D Unet model to denoise each latent plane and add attention layers that operate on the intermediate features of three latent planes following PVDM (Yu et al., 2023). This allows us to effectively model the dependency between planes and use the shared 2D Unet structure across different domains. The training objective of LDM in Eq. 10 is by reparameterization trick $\mu_\varphi(\mathbf{z}_t, t) = \frac{1}{\sqrt{\alpha_t}}(z_t - \frac{\beta_t}{\sqrt{1-\bar{\alpha}_t}} \epsilon_\varphi(z_t, t))$, where $\alpha_t := 1 - \beta_t$ and $\bar{\alpha}_t := \prod_{s=1}^{t} \alpha_s$. We mostly follow techniques proposed in the previous literature. Specifically, we utilize mixed parameterization of score function (Vahdat et al., 2021). We found mixed parameterization beneficial for training the latent space of INRs since we also regularize the posterior distribution towards standard normal distribution in the first stage. Following existing works, we use an exponential moving average (EMA) of the parameters of the LDM network with a 0.9999 decay rate.

## B.3 INFERENCE.

For generation, we use a reverse diffusion process with a fixed number of steps $T = 1000$ (DDPM sampling) for all experiments presented in the main paper unless stated otherwise. However, our model can also leverage recent advanced samplers as discussed in Tab. 12.

# C BASELINES AND EVALUATION METRICS

In this section, we provide a brief overview of baselines and the evaluation metrics employed to assess the performance of the model.

## C.1 2D IMAGE

**Baselines.** CIPS (Anokhin et al., 2021), INR-GAN (Skorokhodov et al., 2021), and ScaleParty (Ntavelis et al., 2022) incorporate adversarial training (Goodfellow et al., 2020) within the StyleGANv2 framework (Karras et al., 2020b). These models utilize fixed single-scale Fourier features for PEs instead of constant input used in StyleGANv2 and modulate the activation function of the network with global latent variables. In particular, ScaleParty (Ntavelis et al., 2022) deploys a CNN-based architecture instead of an MLP and leverages multi-scale training to make the network aware of the generation scale. VaMoH (Koyuncu et al., 2023) employs multiple hyper-generators that generate the weights of INRs with fixed PEs to capture different aspects of signals.

**Fréchet Inception Distance (FID)** (Heusel et al., 2017) is utilized to quantify the dissimilarity between two distributions, typically used in the context of generative models. It operates by comparing the mean and standard deviation of features extracted from the deepest layer of the Inception v3 neural network. In our evaluation, we compute the FID between all available real samples, up to a maximum of 50K, and 50K generated samples, following Karras et al. (2020b). FID helps gauge the quality and diversity of the generated samples by assessing their proximity to the real data distribution. Lower FID scores signify better agreement between the two distributions.

**Improved precision and recall (P&R)** (Kynkäänniemi et al., 2019) measures the expected likelihood of real (fake) samples belonging to the support of fake (real) distribution. They approximate

the support of distribution by constructing K-nearest neighbor hyperspheres around each sample. P&R typically represents fidelity and diversity of generative model, respectively. In our evaluation, we compute the P&R between all available real samples, up to a maximum of 50K, and 50K generated samples.

## C.2  3D SHAPE

**Baselines.**  DPM3D (Luo & Hu, 2021) and PVD construct diffusion processes for point clouds by utilizing point cloud generators. LatentGAN (Chen & Zhang, 2019) and SDF-StyleGAN (Zheng et al., 2022) both make use of a global latent vector to represent global 3D shape features through adversarial training. Specifically, SDF-StyleGAN employs global and local discriminators to generate fine details in the 3D shapes. 3D-LDM (Nam et al., 2022a) learns the global latent vector of SDF through an auto-decoder and trains diffusion models in the latent space. SDF-Diffusion (Shim et al., 2023) introduces a diffusion model applied to voxel-based Signed Distance Fields (SDF) and another diffusion model for patch-wise SDF voxel super-resolution. AutoSDF (Mittal et al., 2022) combines a non-sequential autoregressive prior for 3D shapes conditioned on a text prompt. Diffusion-SDF (Li et al., 2023) encodes point clouds into a voxelized latent space and introduces a voxelized diffusion model for text-guided generation.

**Classification Accuracy**  involves the use of a voxel-based classifier that has been pre-trained to classify objects into 13 different categories within the ShapeNet dataset (Chang et al., 2015), following the approach (Sanghi et al., 2022). It is employed to gauge the semantic authenticity of the generated samples, assessing how well the generated shapes align with the expected categories in ShapeNet.

**CLIP similarity score (CLIP-S)**  employs the pre-trained CLIP model for measuring the correspondence between images and text descriptions by computing the cosine similarity. It evaluates how well the generated shape aligns with the intended text. We follow the evaluation protocol in Li et al. (2023): we render five different views for the generated shape measure CLIP-S for these rendered images and use the highest score obtained for each text description.

**Total mutual difference (TMD).**  In order to measure TMD, we follow the protocols in Li et al. (2023). Specifically, we generate ten different samples for each given text description. Subsequently, we calculate the average Intersection over Union (IoU) score for each generated shape concerning the other 9 shapes. The metric then computes the average IoU score across all text queries. TMD serves as a measure of generation diversity for each specific text query.

## C.3  VIDEO

**Baselines.**  MoCoGAN (Tulyakov et al., 2018) decomposes motion and content in video generation by employing separate generators. DIGAN (Yu et al., 2022) and StyleGAN-V (Skorokhodov et al., 2022) introduce Implicit Neural Representation (INR)-based video generators with computationally efficient discriminators. VideoGPT (Yan et al., 2021) encodes videos into sequences of latent vectors using VQ-VAE and learns autoregressive Transformers. PVDM (Yu et al., 2023) also encodes videos into a projected tri-plane latent space and subsequently learns a latent diffusion model. While our model shares a similar structure with PVDM, it is primarily designed for generating fixed discrete pixels, which limits its adaptability to other domains. There are other recent works, such as VLDM (Blattmann et al., 2023), that utilize diffusion models on the spatio-temporal latent spaces. However, VLDM employs a 3D voxel-based latent space and a 3D-CNN-based network, which can be computationally more intensive. Furthermore, their work relies on a large-scale in-house dataset without open-source code availability, which hinders a fair comparison with our model.

**Fréchet Video Distance (FID)**  is a metric that calculates the Fréchet distance between the feature representations of real and generated videos. To obtain appropriate feature representations, FVD employs a pre-trained Inflated 3D Convnet. In our evaluation, we compute FVD by comparing 2,048 real video samples with 2,048 fake samples, following the preprocessing protocol recommended in Skorokhodov et al. (2022).

Table 11: Generation results on diverse datasets.

| | CIFAR10 $32^2$ | Lsun Churches $128^2$ |
|---|---|---|
| | FID ↓ | FID ↓ |
| <Discrete representation> | | |
| DDPM++(VP) (Song et al., 2021) | 2.47 | - |
| StyleGANv2 (Karras et al., 2020b) | 11.07 | 3.78 |
| LSGM (Vahdat et al., 2021) | 2.10 | - |
| <Continuous representation> | | |
| Domain-specific | | |
| CIPS (Skorokhodov et al., 2021) | 8.62 | 7.38 |
| Domain-agnostic | | |
| GEM (Du et al., 2021) | 23.83 | - |
| DPF (Zhuang et al., 2023) | 15.1 | - |
| **DDMI (Ours)** | 4.53 | 5.12 |

# D  ADDITIONAL RESULTS

## D.1  QUANTITATIVE RESULTS

**CIFAR10 and Lsun Church.**   To demonstrate that DDMI can generalize effectively to datasets with more diverse global structures and classes, we have trained our model on the CIFAR10 and LSUN Church datasets. As shown in Tab. 11, DDMI achieves a significant performance improvement over both domain-specific INR generative model (CIPS) and domain-agnostic INR generative models (GEM and DPF) on both datasets. This result further affirms the efficacy of our design choices, *e.g.*, basie fields generation with D2C-VAE, HDBFs, and CFC, holds for datasets with diverse characteristics.

**Efficient sampling with ODE solver.**   Recent advances in diffusion models, such as sampling via ODE (Ordinary Differential Equation) solvers Song et al. (2021); Vahdat et al. (2021); Karras et al. (2022), have shown promising results in reducing sampling time. We use an RK45 ODE solver, instead of ancestral sampling, similar to  Vahdat et al. (2021).

By increasing the ODE solver error tolerances, we generate samples with lower NFEs (number of function evaluations). In Tab. 12, we report FID scores on AFHQv2 Cat and Dog datasets for three cases: 1) T=1000, 2) ODE solver error tolerances of $10^{-3}$, and 3) ODE solver error tolerances of $10^{-5}$. The results demonstrate that our DDMI framework performs efficient sampling while still achieving satisfactory performance (see Fig. 16 and 17 for qualitative results).

Table 12: NFE vs. FID.

| Avg. NFE | AFHQv2 Cat FID | AFHQv2 Dog FID |
|---|---|---|
| 1000 | 4.27 | 8.54 |
| 50 | 5.32 | 11.04 |
| 25 | 5.93 | 14.23 |

## D.2  QUALITATIVE RESULTS

**Visualization of the nearest neighbor samples.**   In order to confirm that our model is not overfitting to the training dataset, we employ a visualization technique that displays the nearest training samples to the generated samples in the VGG feature space. This visualization is presented in Fig. 13.

**Arbitrary-scale 2D image synthesis.**   Fig. 8, 9, and 10 show the generated 2D images at arbitrary resolutions, including $256^2$, $384^2$, and $512^2$. These results demonstrate the ability of our model to generate continuous representations with high quality.

**3D shape generation.**   In Fig 12, we compare the qualitative results on 3D shape with two recent baselines: GASP (Dupont et al., 2022b) (domain-agnostic) and SDF-StyleGAN (Zheng et al., 2022) (domain-specific). The result demonstrates that our method generates more sophisticated details while maintaining smooth surfaces compared to the baselines. Moreover, Fig. 11 shows that our DDMI generates diverse and high-fidelity 3D shapes across various object categories: airplane, car, chair, desk, gun, lamp, ship, *etc*.

**Video generation.** Fig. 14 presents video generation results on the SkyTimelapse dataset. The results demonstrate the capability of our model to generate visually appealing and coherent sequences of images.

**Neural radiance fields generation.** Fig. 15 displays the NeRF generation results of our model on the SRN car dataset in comparison to Functa.

## E BROADER IMPACTS

The development of generative models with controllability across diverse data domains has been a long-standing objective in the field. Our proposed generative model, DDMI, represents a significant advancement in the realm of INR generative models, offering high-fidelity generation and effective applicability to a wide range of data domains. Moreover, one key strength of DDMI is its capability to generate samples at arbitrary scales, providing enhanced controllability that can be leveraged in conjunction with existing conditional generation techniques Rombach et al. (2022). This opens up exciting possibilities for tasks like text-guided super-resolution video generation, where the fine-grained control offered by DDMI can yield compelling results. By enabling controllable generation across different data modalities, DDMI has the potential to drive further advancements in creative applications, and other domains that rely on generative modeling.

However, it is important to acknowledge potential concerns that arise with the deployment of generative models Tinsley et al. (2021), including DDMI. Like other generative models, there is a risk of revealing private or sensitive information present in the data. Additionally, generative models may exhibit biases Esser et al. (2020) that are present in the dataset used for training. Addressing these concerns and ensuring the ethical and responsible deployment of generative models is crucial to mitigate potential negative impacts and promote fair and inclusive use of the technology.

## F LIMITATIONS

Domain-agnostic INR generation has shown promising results in a wide range of applications with great flexibility. The proposed method enhances the expressive power of INR generation via asymmetric variational autoencoder (VAE) connecting the discrete data space and the continuous function space. One caveat of the proposed method is the relatively long generation time. This is one well-known common disadvantage of diffusion models. In Sec. D.1, we studied advanced solvers to speed up the generation process but it is not satisfactory. With a small number of function evaluations (NFEs), the generation suffers from quality degradation. More efficient generation schemes leveraging compact latent spaces will be an interesting future direction.

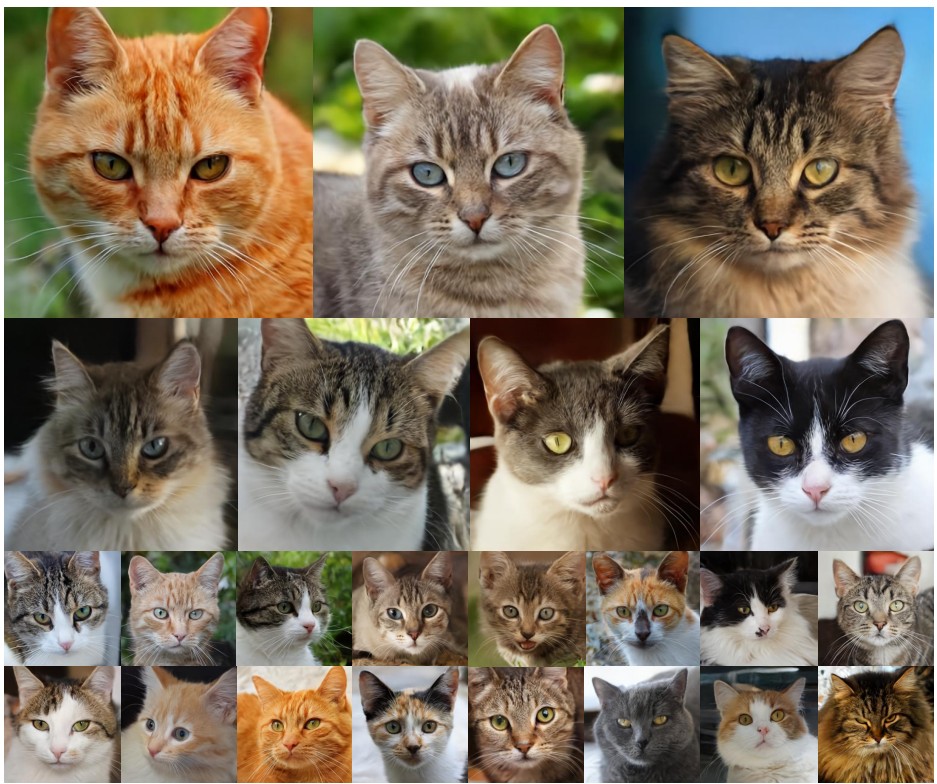

Figure 8: Uncurated samples on AFHQv2 Cat for various resolutions generated from single DDMI model.

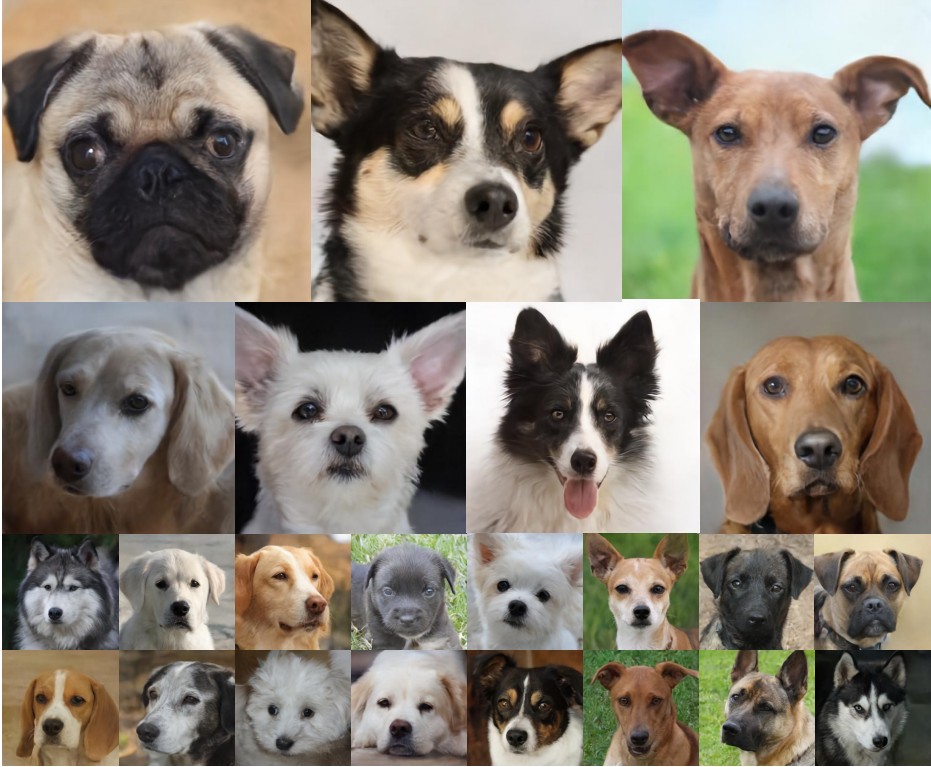

Figure 9: Uncurated samples on AFHQv2 Dog for various resolutions generated from single DDMI model.

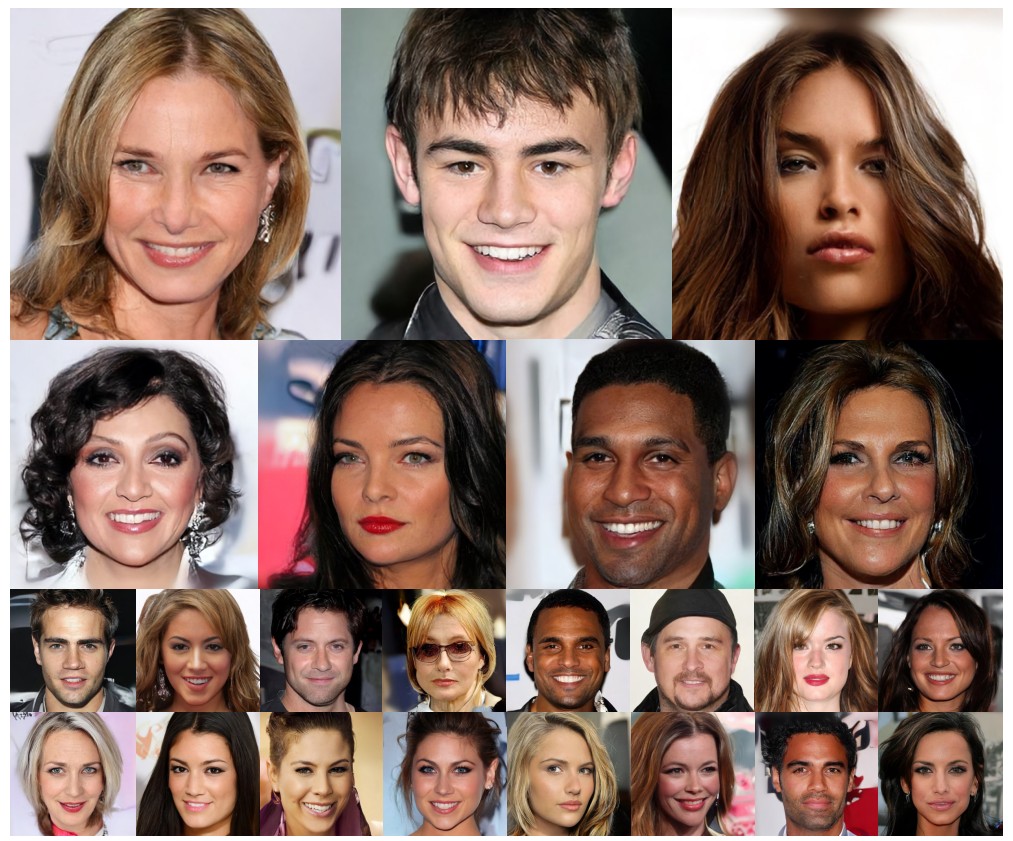

Figure 10: Uncurated samples on CelebA-HQ for various resolutions generated from single DDMI model.

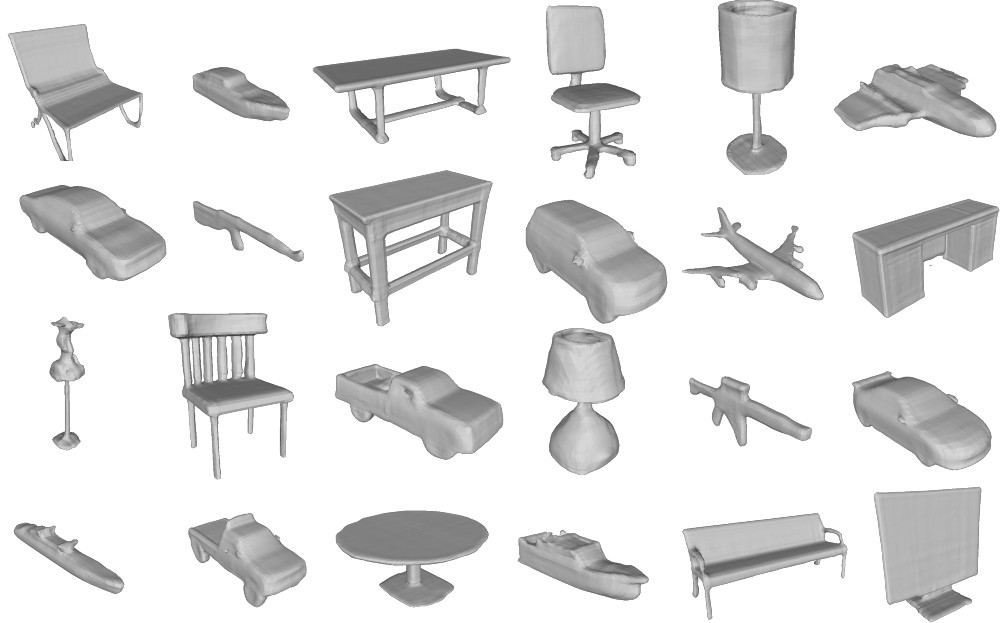

Figure 11: Uncurated samples on ShapeNet generated from DDMI model.

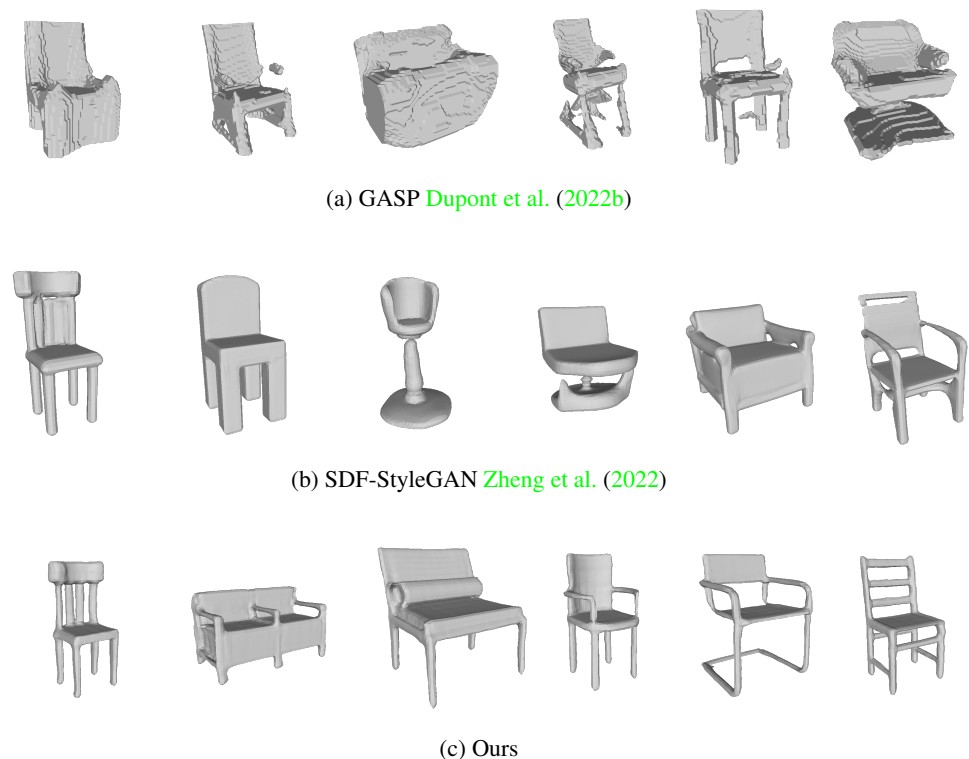

(a) GASP Dupont et al. (2022b)

(b) SDF-StyleGAN Zheng et al. (2022)

(c) Ours

Figure 12: **Qualitative comparison on 3D INR generation.** We compare GASP (domain-agnostic), SDF-StyleGAN (domain-specific), and DDMI on 3D shapes.

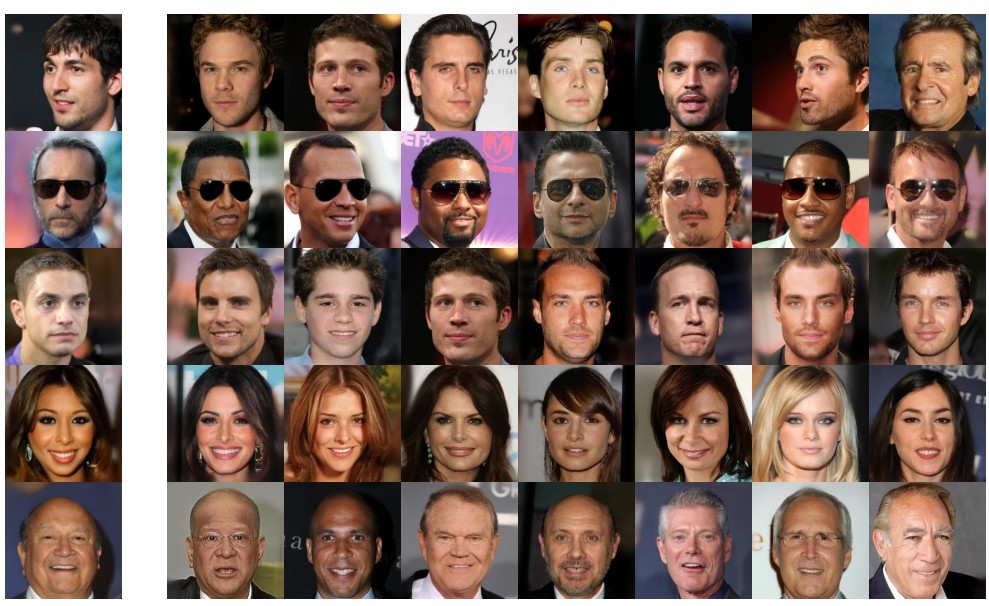

Figure 13: Nearest neighbors of our CelebA-HQ model, computed in the features of VGG16. The leftmost sample is from our model, and the remaining samples in each row are its 7 nearest neighbors.

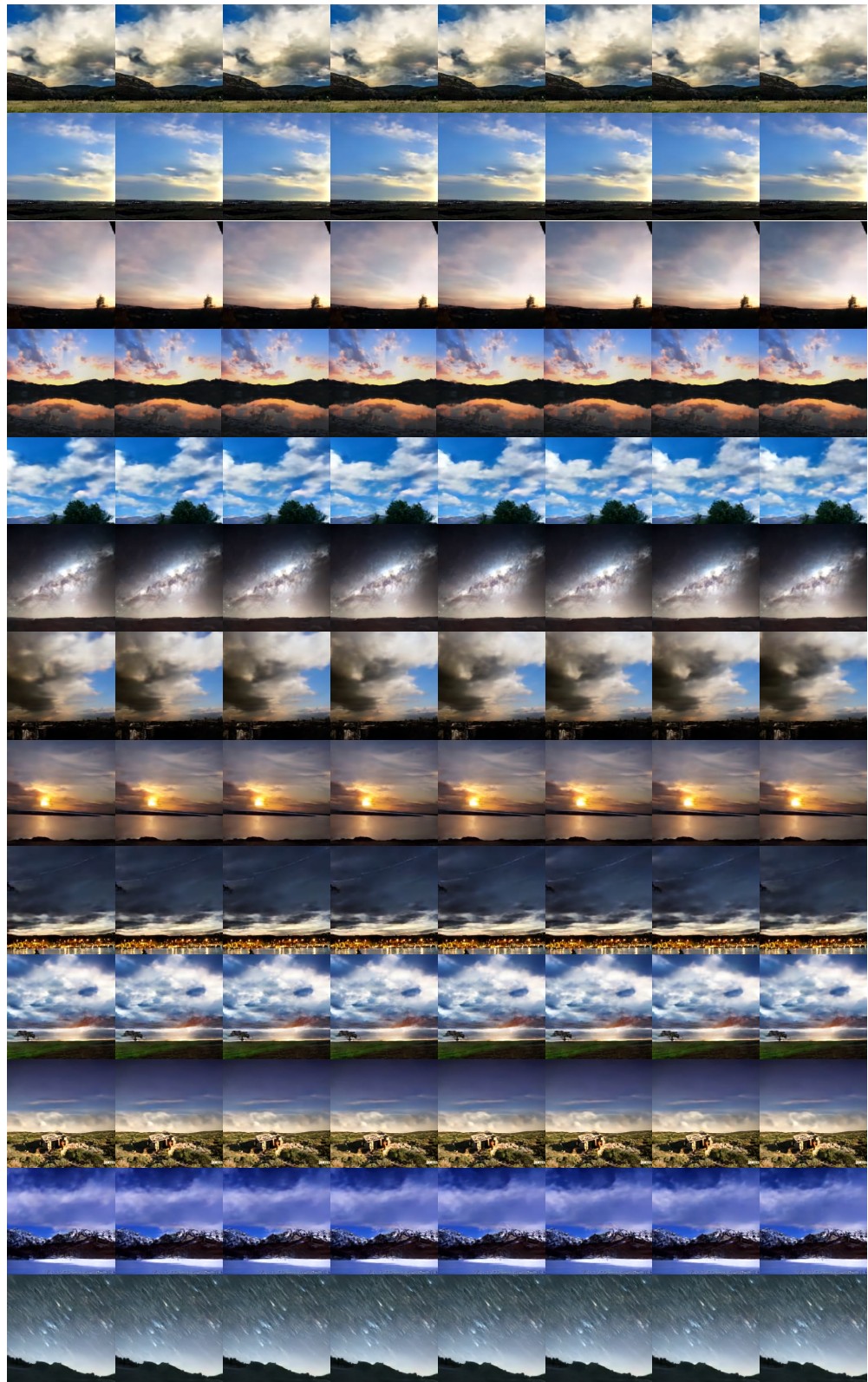

Figure 14: Uncurated samples on SkyTimelapse generated from DDMI.

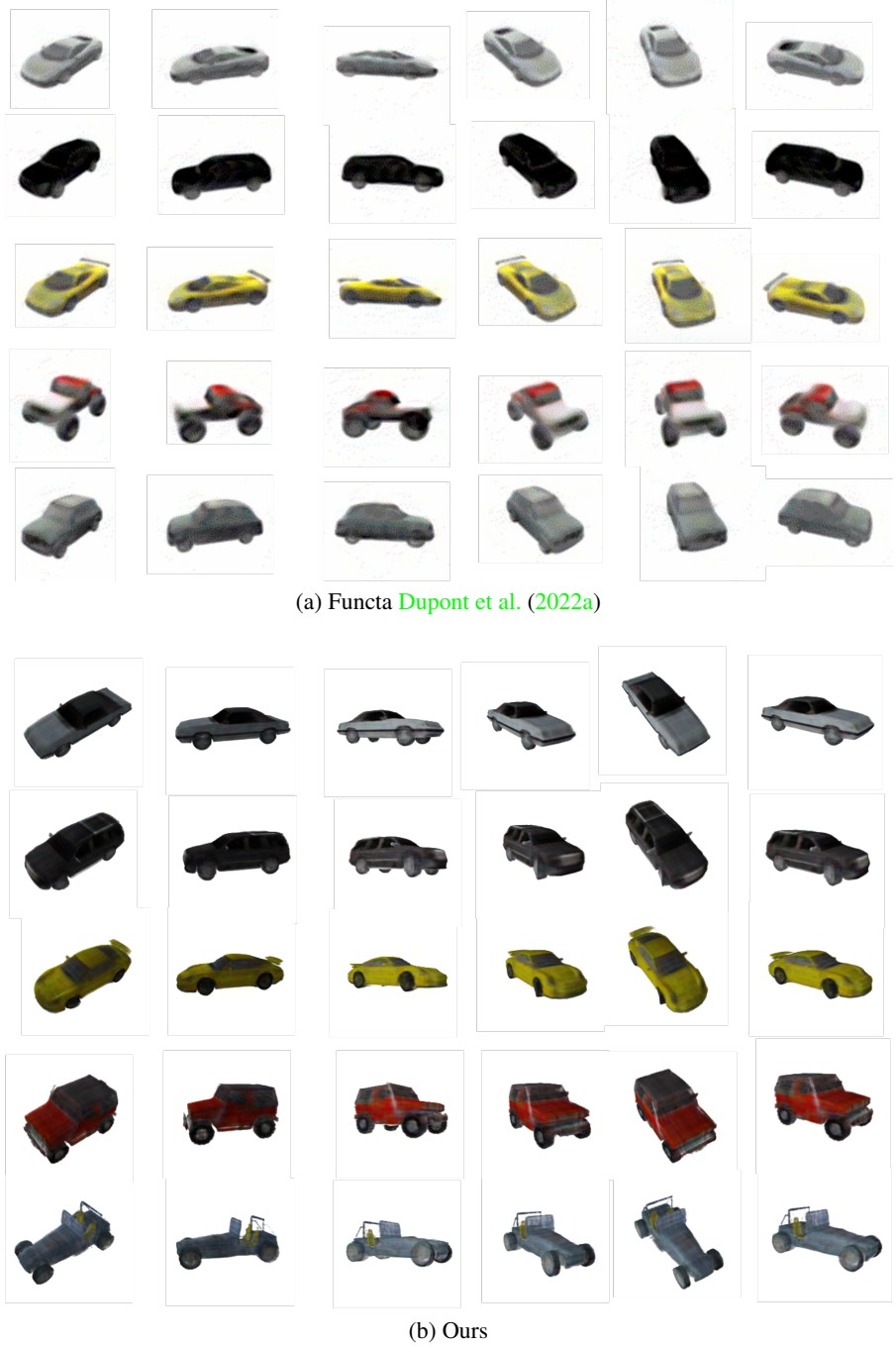

Figure 15: **Qualitative comparison on NeRF generation.** Our model generates sophisticated details and vivid color texture while maintaining smooth surfaces. In contrast, Functa exhibits limitation in capturing intricate details and tends to produce blurry texture.

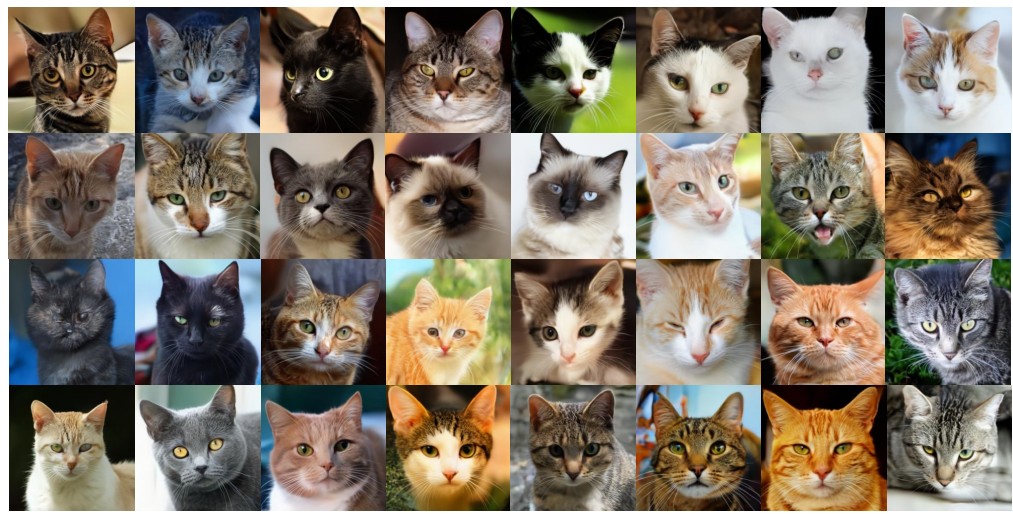

Figure 16: Uncurated samples on AFHQv2 Cat generated from DDMI model using ODE solver with $10^{-5}$ tolerance (Avg. NFE=25).

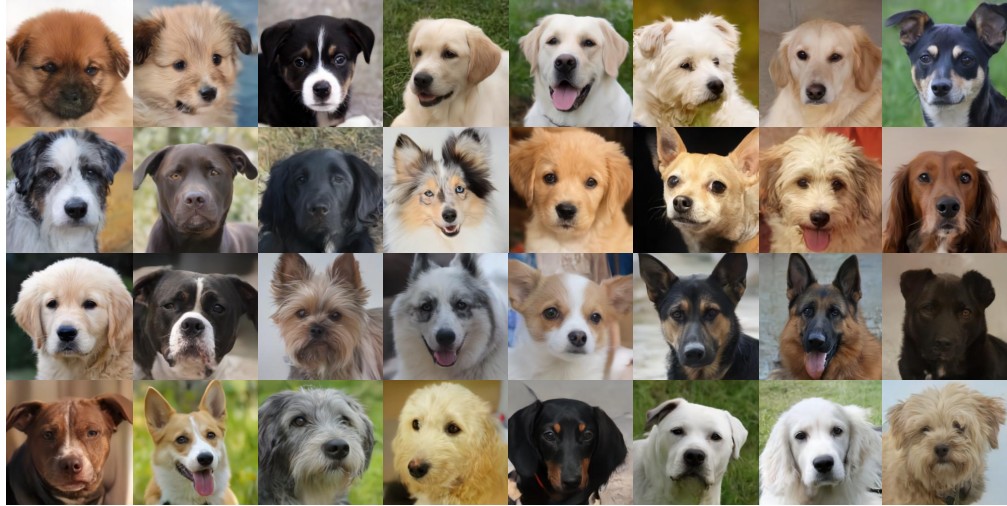

Figure 17: Uncurated samples on AFHQv2 Dog generated from DDMI model using ODE solver with $10^{-3}$ tolerance (Avg. NFE=50).

