# OpenReview forum: "DDMI: Domain-agnostic Latent Diffusion Models for Synthesizing High-Quality Implicit Neural Representations"
_ICLR.cc/2024/Conference — ICLR 2024 poster_

### Official Review · Reviewer_9btm · 2023-10-30

**Soundness:** 4 excellent
**Presentation:** 3 good
**Contribution:** 3 good
**Rating:** 6
**Confidence:** 5

**Summary:**

The paper presents a probabilistic model for functions using INRs. The main features are:

- A VAE projects the discrete data input X into a latent space.
- An adaptive PE module replaces hyper network that generates weights. The VAE decoder generates a hierarchy of basis fields that account for the multi-scale properties of the data to be generated.
- This hierarchy of PE is introduced sequentially into the reconstructing MLP.

The DDMI model presents robust performance across a variety of experiments.

**Strengths:**

- The paper is well-written and clear
- Experimental setting demonstrates a robust performance compared to other methods
- Methodology is rigorous and conclusions are supported by evidence.

**Weaknesses:**

To my understanding, the present work presents a well-designed combination of features and methods already present in the state-of-the-art:

- The use of signal fields and adaptive PE in implicit models has been reported by Zhuang et al. (ICLR 2023).
- A VAE approach for probabilistic INRs has been recently published by Koyuncu et al. (ICML 2023): "Variational Mixture of HyperGenerators for Learning Distributions Over Functions." This paper is not even mentioned in the present work, and it is a must for the authors to include it, not only in the literature review but also in the experimental setting. Authors also consider using various hypergenerators to capture different data scales and modalities in that work.
- I do not think the paper also compares with "Implicit Diffusion Models for Continuous Super-Resolution" (CVPR 2023), in terms of tools and results obtained. Comparisons with this paper are important too.
- Using a generative latent diffusion model directly follows the idea of Rombach et al. (CVPR 2022).
- Hierarchies of multi-scale PE have already been developed in the literature. The paper "Composite Slice Transformer: An Efficient Transformer with Composition of Multi-Scale Multi-Range Attentions" (ICLR 2023) is a good reference.

While combining existing ideas into a robust model is not a demerit of the paper per se, the paper's contributions are limited.

Also, when explaining the interpolation and the basis functions, more details are needed since these concepts are hard to understand the way they are written.

**Questions:**

Did you try to train a more powerful VAE latent space instead of training a latent diffusion model? I'm just wondering how important is to incorporate the diffusion model. It seems to me that the DDMI gains mostly come from using the adaptive hierarchical PE.

I did not see many details about the posterior distribution used in the training of the D2C-VAE. Can you please introduce more details about this point?

What is the dimension of the latent space z? Have you studied the model's performance as this dimension varies?

---

> ### Author Response · Authors · 2023-11-15
> **Reply to Reviewer 9btm**
>
> We thank your detailed and thoughtful feedback for our work. We will address the concerns below.
>
> 1. [W1] **Lack of references & comparison with references.**
>
>     We appreciate your reference to the recent papers that we have overlooked. We will include all the papers in the manuscript.
>
>     - Regarding the paper ICML 2023.
>
>         Indeed, their work also employs a VAE framework for modeling INRs and utilizes multiple hyper-generators to capture different aspects of signals. However, they still model the weight space of functions with fixed PEs, similar to previous generative models for INRs.
>
>         In our paper, we have proposed generating PEs instead of generating weight space, and have quantitatively shown that it is more effective in generating high-fidelity signals. This is also supported by quantitative evidence presented in the table below, where our model outperforms the VaMoH approach in terms of signal fidelity on the CelebA 64x64 dataset.
>
>         |  | CelebA 64x64 |
>         | --- | --- |
>         | VaMoH | 66.27 |
>         | Ours | 9.74 |
>     - Regarding the paper CVPR 2023.
>
>         We want to stress that scale-free image generation demonstrated in our paper differs from the super-resolution task. (CVPR 2023) proposes a specialized model architecture and methodology tailored for super-resolution tasks, which cannot generate diverse images like ours. Therefore, it is not feasible to make a fair comparison with our work.
>
> $$\\\\$$
>
> 2. [W2] **More details regarding interpolation/basis functions.**
>
>     We will improve the manuscript accordingly within the rebuttal period and notify you after the update.
>
> $$\\\\$$
>
> 3. [Q1] **Did you try to train a more powerful VAE latent space instead of training a latent diffusion model?**
>
>     During the early stages of our research, we experimented with training more powerful VAEs, including hierarchical VAEs [A]. Nevertheless, we observed that even these advanced VAEs encounter challenges in producing high-quality INRs, primarily due to issues with the 'prior hole' phenomenon. This limitation was found to be less significant in latent diffusion models, which is why we pursued that direction.
>
>    ---
>    [A] Vahdat and Kautz, NVAE: A Deep Hierarchical Variational Autoencoder, NeurIPS 2020
>
> $$\\\\$$
>
> 4. [Q2] **Posterior distribution used in D2C-VAE.**
>
>     We assumed multivariate normal distribution for posterior $q_\phi(\mathbf{z}|\mathbf{x})$ in all domains.
>
> $$\\\\$$
>
> 5. [Q3] **Dimension of latent space.**
>
>     The dimensions of the latent space $\textbf{z}$ are detailed in the supplementary materials, specifically in Table 9. We found that reducing the dimensionality of the latent space compromises sample quality to some extent, but it also results in a more efficient computational budget. This trade-off is consistent with observations from LDMs [B].
>
>     ---
>     [B] Rombach et al., High-Resolution Image Synthesis with Latent Diffusion Models, CVPR 2022

---

> > ### Comment · Reviewer_9btm · 2023-11-19
> >
> > Dear authors,
> >
> > Thanks a lot for your promt response. The comparison with the ICML’23 paper is convincing and the difference w.r.t. the super-resolution paper in CPVR 23 clear. I strongly suggest the authors to include in the supplementary part the results when a hierarchical VAE is used instead of latent diffusion prior.
> >
> > If more details are included when explaining the use of basis/interpolation functions, then I’m willing to increase my rating.

---

> ### Author Response · Authors · 2023-11-20
> **Reply to Reviewer 9btm (2)**
>
> Thank you for your reply and support.
> As you suggested, we have updated the manuscript to provide more details about the basis/interpolation function.
> In this paper, we have defined basis fields as a set of dense grids (vector fields) consisting of generated basis vectors to be interpolated for generating a positional embedding given coordinates. See Section 3.1 & Section 4 in the main paper and Appendix A & Table 9 in the Appendix, and inform us whether the revised explanation sufficiently addresses any previous concerns.

---

### Official Review · Reviewer_KC2J · 2023-10-31

**Soundness:** 3 good
**Presentation:** 3 good
**Contribution:** 3 good
**Rating:** 6
**Confidence:** 4

**Summary:**

This work proposes a new INR generative model called Domain-agnostic Latent Diffusion Model (DDMI) that can generate continuous signals in various domains. DDMI consists of the D2C-VAE and a diffusion model in its latent space. The D2C-VAE maps discrete data into a latent space and then generates hierarchically decomposed basis fields for adaptive PEs, shifting the primary expressive capacity from MLP to PE. Coarse-to-Fine Conditioning is applied to return the signal value for the queried coordinate, further improving the performance of generating INR.

**Strengths:**

1. **The proposed method is straightforward and effective.** The proposed position embedding modeling and hierarchical decoding are straightforward, yet they are versatile and effective across seven benchmark datasets.

2. **Experiments are thorough.** Experiments across four modalities and ablation studies including the proposed two modules are conducted, indicating the superior performance and versatility of DDMI.

3. **Analysis of the decomposition of HDBFs is intriguing.** The investigation strategy and visualizations of results in spatial and spectral domains provide insightful information.

**Weaknesses:**

**Some expressions are confusing and unclear.**
For instance, when it comes to the tri-plane grid features shown in Figure 2, the input discrete data includes images, 3D shapes and videos. However, the tri-plane is only applicable to 3D shapes and videos, and not to 2D images, which use a 2D plane as their latent space type.
Additionally, there should be more information provided about D2C-VAE in the main paper, such as the representation and dimensionality of latent variables and basis fields, rather than including it only in the appendix. It becomes difficult to follow the information without having to read the appendix.
Lastly, It's also unclear how to project sparse point clouds to be processed by a 2D CNN.

**Questions:**

1. **Why use the occupancy function to represent a 3D shape instead of the SDF?** Many compared baselines learn a generative model on 3D INR by modeling the distribution of SDF, but this work adopts the occupancy function. How will using SDF as INR affect the proposed method's performance? Can the difference in 3D shape representation cause unfair comparisons?

2. **How to project sparse point clouds to be processed by a 2D CNN?** It is mentioned in Appendix A that an orthographic projection is performed on the PointNet local pooling features to three canonical 2D planes, and a shared 2D CNN encoder is used to obtain tri-plane latent features. However, the output of a PointNet encoder with a local pooling operation is still a sparse "point cloud" in the shape of `BxNxD`, and the adjoined arrays in the tensor are not actually adjoined spatially. How does a 2D CNN perform on the sparse input?

3. **Will the code be publicly available?** Releasing the code is not mentioned in the paper, but it would be helpful to facilitate future related research and provide much more details about the method.

---

> ### Author Response · Authors · 2023-11-15
> **Reply to Reviewer KC2J**
>
> Thank you for the constructive feedback and suggestions. We will address the mentioned questions below.
>
> 1. **[W1] Unclear expressions + $\alpha$**
>
>     We will address the issues outlined below:
>
>     - Tri-plane in the figure
>         - You’re right, for the case of 2D images, the latent type would be a single plane, rather than a tri-plane. We will clearly clarify this in the main figure of the manuscript.
>     - More implementation details in the main paper
>         - As you suggested, we will add implementation details from appendix into the main paper for better readability.
>     - Projection of sparse point clouds
>
>         Below, we will elaborate the orthographic projection process from Conv-ONet [A]:
>
>         1. **Feature Extraction**: PointNet processes the input point cloud to produce feature tensor of shape `BxNxD` .
>         2. **Projection**: For each point, we apply an orthographic projection to map its feature onto three canonical planes (xy, yz, zx) of size `HxW`. Features projecting onto the same pixel are aggregated using average pooling, resulting in three distinct 2D grid features, each with the dimensionality `HxWxD`.
>
>         Thus, the resulting 2D grids can be processed 2D CNN. We refer **Figure 2(a)** of Conv-ONet for better understanding.
>
>
>     We appreciate your considerate comments. We will edit the manuscript accordingly within the rebuttal period and notify you after the update.
>
>
>
>
>     ---
>
>     [A] Peng et al., Convolutional Occupancy Networks, ECCV 2020
>
> $$\\\\$$
>
> 2. **[Q1] Why occupancy function over SDF?**
>
>     We are aware of the superior expressiveness power of SDF over occupancy functions. However, we favored occupancy functions over SDF for 3D INR due to the significant reduction in preprocessing time and dataset size. For instance, the SDF representation of the ShapeNetV2-chair dataset occupies roughly 620GB, whereas the occupancy function representation requires only about 12GB. Surprisingly, the results suggest that our method with occupancy functions yields shape quality that is comparable or superior to SDF-based methods, both qualitatively and quantitatively. This observation was consistent despite the inherent differences in the representations.
>
> $\\\\$
>
> 3. **[Q3] Code release**
>
>     Of course, we will release our code in the final version.

---

> > ### Comment · Reviewer_KC2J · 2023-11-16
> >
> > Thank you for the reply which solved most of my concerns. I also read other reviews and the corresponding replies. Generally, I think this is a good paper because (1) it is the first work to model positional embeddings to generate INRs, (2) multi-scaled strategies are proposed to promote the performance, and (3) experiments are extensive, conducted in many domains and demonstrate superior performance. I'm considering increasing my rating in the reviewer discussion period.
> >
> > However, I still have some suggestions for this work.
> >
> > - Basically, the key contribution of this work is D2C-VAE, which provides a continuous positional embedding field for INRs. However, the paper prominently features latent diffusion models (LDM), even putting the term in the title, which appears that the emphasis is somewhat disproportionate. From the perspective of LDM, the contribution of this work is incremental, as it's just a direct application. In fact, any generative models can be applied to model the VAE latent space, as Dupont et. al (2022) did, using NSF and DDPM to model functa. Therefore, I recommend de-emphasizing LDM, and the authors could conduct experiments to elucidate the reasons for selecting LDM, instead of other generative models.
> >
> > - The main paper dedicates a substantial portion to introducing the preliminaries about VAE and diffusion models (Sec. 3.2), but leaves the details about the key contribution, i.e., D2C-VAE, to the appendix. It makes the paper hard to follow without checking the appendix.

---

> ### Author Response · Authors · 2023-11-20
> **Reply to Reviewer KC2J (2)**
>
> - **Emphasizing other contributions rather than LDM.**
>
>    Thank you for your valuable feedback and suggestions. In this paper, we proposed a new meta-architecture for generating INRs given discrete inputs. Our architecture uses a latent diffusion model in the latent space of an asymmetric VAE that learns a joint latent space that bridges discrete data to continuous functions. The LDM in our title/model name describes the overall architecture of the proposed method similar to other models’ names in the literature [B,C,D].
>
>     Indeed, reviewer KC2J raised a great point that our contributions are not limited to diffusion models. Our main contributions (D2C-VAE, HDBFs, CFC) are definitely applicable to other generative models, such as Normalizing Flows and GANs. As suggested, we will provide additional experiments with other generative models other than latent diffusion models to demonstrate the extensionality of our contributions.
>
>     ---
>     [B] Liu et al., AudioLDM: Text-to-Audio Generation with Latent Diffusion Models, ICML 2023
>
>     [C] Koo et al., SALAD: Part-Level Latent Diffusion for 3D Shape Generation and Manipulation, ICCV 2023
>
>     [D] Stan et al., LDM3D: Latent Diffusion Model for 3D, CVPRW 2023
>
> $$\\\\$$
>
> - **Improving the presentation of D2C-VAE + $\alpha$.**
>
>    As suggested, we have updated the manuscript (see sections 3.1, 3.2, 4, and Appendix A). Note that we reflect on your previous feedback in the current manuscript as well (clarification on the main figure, details on projection of point clouds).

---

### Official Review · Reviewer_qBmp · 2023-10-31

**Soundness:** 2 fair
**Presentation:** 3 good
**Contribution:** 3 good
**Rating:** 6
**Confidence:** 3

**Summary:**

This paper proposes a domain-agnostic latent diffusion models for generating implicit neural representations across three domains: images, videos, and 3D shapes. The main novelty is that they let the latent variables generating adaptive basis fields for different examples, and learn a single rendering function afterwards. Multi-resolution basis fields have been used to further boost the performance. Extensive experiments have been performed on those three domains to showcase that the method works better than other domain-agnostic INR approaches.

**Strengths:**

The paper proposed to instead of learning to generate different rendering functions, learn to generate different basis fields, which seems to be a more principled way. The effectiveness has been justifed by the extensive experiments and comparison with baseline approaches. The multiscale structure of generating basis fields seems to help generating fine-grained details for high-resolution rendering.

**Weaknesses:**

- I'm a bit concerned about the soundness of D2C-VAE the paper proposed. Specifically, how to justify eq. (2) is a valid training objective, does it correspond to an ELBO of a log-likelihood function? It'd be better to formulate the problem and training objective in a more principled and statistically sound way, e.g., by expressing everything in the function space.

- It'd be nicer to have some intuitive explanation on the reason why modeling PE works better than modeling the the rendering function $\omega$.

- All experimental results are shown on relatively easy datasets on images, 3d and videos.Would be curious to see how this approach work with more challenging and scaled up datasets on various domain.

**Questions:**

Please see the comments above.

---

> ### Author Response · Authors · 2023-11-15
> **Reply to Reviewer qBmp**
>
> We appreciate the reviewer for the valuable feedback. We will address the raised questions below.
>
> 1. **[W1] Regarding the soundness of eq.(2).**
>
>     We would like to clarify the rationale behind our D2C-VAE. The objective of D2C-VAE is to maximize the ELBO of the log-likelihood of a continuous function $\boldsymbol{\omega}$ with the discrete data $\textbf{x}$. It is articulated below based on the standard variational inference principles: $$ \begin{align} \log p(\boldsymbol{\omega})  & = \log \int p_{\psi, \pi_\theta}(\boldsymbol{\omega}| \mathbf{z}) \cdot p(\mathbf{z}) d\mathbf{z} \\\\ & = \log \int \frac{p_{\psi, \pi_\theta}(\boldsymbol{\omega}|\mathbf{z})}{q_\phi(\mathbf{z}|\mathbf{x})} \cdot q_\phi(\mathbf{z}|\mathbf{x}) \cdot p(\mathbf{z}) d\mathbf{z} \\\\ & \geq \int \log \left(\frac{p_{\psi, \pi_\theta}(\boldsymbol{\omega}|\mathbf{z})}{q_\phi(\mathbf{z}|\mathbf{x})} \cdot p(\mathbf{z}) \right) \cdot q_\phi(\mathbf{z}|\mathbf{x}) \ d\mathbf{z} \\\\ & = \int q_\phi(\mathbf{z}|\mathbf{x}) \cdot \left( \log p_{\psi, \pi_\theta}(\boldsymbol{\omega}|\mathbf{z}) - \log \left( \frac{q_\phi(\mathbf{z}|\mathbf{x})}{p(\mathbf{z})} \right) \right) \\\\ & = E_{q_\phi(\mathbf{z}|\mathbf{x})}\left[\log p_{\psi, \pi_\theta}(\boldsymbol{\omega}|\mathbf{z}) \right] - D_{KL}(q_\phi(\mathbf{z}|\mathbf{x}) || p(\mathbf{z})) \end{align} $$
>
>    where the inequality in the third line is by Jensen's inequality and the last line corresponds to Equation 2 in our paper. However, we do not have direct observations to $\boldsymbol{\omega}$, while we have access to discrete data $\textbf{x}=\boldsymbol{\omega}(\mathbf{c})$, we approximate $p_{\psi, \pi_\theta}(\boldsymbol{\omega}|\mathbf{z})$ as
>
>    $$\begin{align}
>     p_{\psi, \pi_\theta}(\boldsymbol{\omega}|\mathbf{z}) & \approx p_{\psi, \pi_\theta}(\boldsymbol{\omega(\mathbf{c})}|\mathbf{z})
>     \\\\
>     & = \prod_{c \in \mathbf{c}} p_{\psi, \pi_\theta}(\boldsymbol{\omega}(c)|\mathbf{z}), \end{align} $$ where we assume the coordinate-wise independence. This leads to the approximation in Equation 3 of our paper. Assuming the likelihood as a multivariate normal distribution with isotropic covariance ($\sigma$),
>
>    $$\begin{align}
>      p_{\psi, \pi_\theta}(\boldsymbol{\omega(\mathbf{c})}|\mathbf{z}) = \prod_{c \in \mathbf{c}} p_{\psi, \pi_\theta}(\boldsymbol{\omega}(c)|\mathbf{z}) =\prod_{c \in \mathbf{c}} \frac{1}{Z} \exp \left( - \frac{||\boldsymbol{\omega}(c) - \hat{\boldsymbol{\omega}(c)||^2_2}}{2\sigma^2} \right) \end{align} $$  where $Z$ is a normalizing constant and $\hat{\boldsymbol{\omega}}(\mathbf{c}) = \pi_\theta(\gamma(\textbf{c}, D_\psi(\mathbf{z})))$. We will update this in the appendix.
>
> $$\\\\$$
>
> 2. **[W2] Intuition behind PE generation instead of weight generation.**
>    We will provide the intuition of PE generation. Recent INR studies in the literature have shown the significance of positional encoding (PE) to achieve **high-quality** representations. Parametric PEs, such as hash grids and orthogonal planes with **learnable** parameters [A,B,C], exhibit a superior expressive capacity compared to fixed PE (e.g., Fourier feature mappings [D]). Thus, we favor learning to generate PE over MLP weights based on the outstanding expressive power of parametric PEs. In EG3D [E], they have demonstrated the generation of high-fidelity 3D-aware 2D images by generating tri-plane PEs based on GAN. Unlike the prior work, we propose PE generation with the latent **diffusion model**, enhancing it by our novel modules such as **multi-resolution framework (HDBF)** and **coarse-to-fine conditioning (CFC)**.
>
>    ---
>    [A] Muller et al., Instant neural graphics primitives with a multi resolution hash encoding, ACM ToG 2022
>
>    [B] Cao et al., Hexplane: A fast representation for dynamic scenes, CVPR 2023
>
>    [C] Fridovich-Keil et al., K-planes: Explicit radiance fields in space, time, and appearance, CVPR 2023
>
>    [D] Mildenhall et al., NeRF: Representing Scenes as Neural Radiance Fields for View Synthesis, ECCV 2020
>
>    [E] Chan et. al., Efficient geometry-aware 3d generative adversarial networks, CVPR 2022
>
> $$\\\\$$
>
> 3. **[W3] Experiments with more complex datasets.**
>
>     Due to limited resources, we could not conduct experiments with more complex, large-scale datasets like Objaverse or ImageNet. Note that we have evaluated our method on a relatively larger dataset like LSUN church (126K samples ) as mentioned in the appendix. We are committed to extending our evaluation to more challenging datasets in the future.

---

> > ### Author Response · Authors · 2023-11-20
> > **Reply to Reviewer qBmp (2)**
> >
> > We would like to notify the reviewer that we have updated the manuscript to clarify the soundness of equation 2 (see section 3.2).
> >
> > Also, to support the intuition for preferring PE generation over weight generation, we refer to the above section (see the reply for Reviewer 6uL5), where we provide the comparison between weight vs PE generation (without HDBFs and CFC). We hope this additional experiment can convince the superiority of PE generation.

---

### Official Review · Reviewer_6uL5 · 2023-11-04

**Soundness:** 3 good
**Presentation:** 3 good
**Contribution:** 2 fair
**Rating:** 6
**Confidence:** 4

**Summary:**

This work presents an improved framework for designing a domain agnostic latent diffusion architecture that can right away model the distributions over any class of continuous implicit neural functions (INR) without any architectural modifications.
To that extent, the existing works attempts to generates the weights of MLP that parameterize INRs while keeping the input positional embeddings (PEs) fixed.
However, the recent works on INRs demonstrates that, careful designs of PEs instead of MLPs achieves better efficiency and effectiveness in representing INRs.
Motivated by this development, this work proposes DDMI that generates sample-specific adaptive PEs while keeping the weights of MLP fixed.
Specifically, they develop D2C-VAE whose encoder maps explicits discrete space signals to latent space and its decoder maps latent space to continuous basis fields.
Apart from LDM for D2C mapping, DDMI hierarchically decomposes basis fields (HDBFs) into multiple scales while the MLP progressively performs coarse to fine conditioning (CFC) using these multi-scale PE from HDBFs.
Extensive experiments across different input modalities shows DDMI outperforming recent SoA on INR generative models.

**Strengths:**

Quality and Clarity
The paper is well-written and easy to follow. Many quantitative results, ablation studies, and implementation details with hyper-parameters are discussed in the supplementary. These served as helpful guide.

Originality and Significance
1. It is the first work to combine the power of latent diffusion model with input adaptive positional embeddings for generating implicit neural representations.
2. Same architecture is used for generating arbitrary continuous INRs using discrete inputs from various domains that includes images, videos, 3D shapes and NeRFs.
3. Overall, better qualitative and quantitative results when compared to SoA discrete generative models, domain-specific INR generative models and domain-agnostic generative models.
4. Due to explicit tuning of multiple scales, DDMI is capable of generating signals of multiple resolutions without any loss of signal fidelity.

**Weaknesses:**

1. Limited novelty. I see this work as a natural extension to other works on domain-agnostic LDMs for INRs such as Functa (Dupont et. al, 2022a) and DPF (Zhuang et. al, 2023). The only difference between DDMI and Functa is replacing dynamic MLP weights with dynamic positional embeddings.
2. DDMI made three changes w.r.t. Functa -- dynamic MLP weights -> dynamic PEs, HDBFs, and CFC. It is still unclear how much improvement was achieved by making a shift from dynamic MLP to dynamic PEs. The ablation only covers the relative effects of HDFCs and CFC.
3. Scalability -- The work claims that the prior works are not scalable as dynamically modifying MLP weights limit its expressive power. However, DDMI has not demonstrated any scalability. Most of the results are either on single or few class dataset, while the number of samples are in the order of 10K. Currently, it seems to me that semantic class specific INR generative model is being fit. True scalability to larger community is model's ability to generate high quality samples from diverse set of classes. Consider training on Objaverse scale of dataset.
4. Significance -- Given that almost all high-fidelity LDM models are obtained by domain-specific LDMs, I fail to understand the real value behind creating domain-agnostic architecture. It make sense to anyone, if priors from multiple domains are reused which is possible only if DDMI was trained by sampling input discrete data across domains.
5. Domain Agnostic ? It is not clear how the presented DDMI is domain-agnostic. I presume the input and output of LDMs will remain the same when trained for signals from each domain. However, the implementation details in appendix suggests that dataset-specific architecture was deployed with LDM input/output also changing likewise.

**Questions:**

Please provide clarity on points raised under weaknesses section.

---

> ### Author Response · Authors · 2023-11-15
> **Reply to Reviewer 6uL5**
>
> We appreciate your thorough and insightful feedback. We will address the raised questions below.
>
> 1. **[W1, W2] Limited novelty & why dynamic PE?**
>
>     Recent INR studies in the literature have evidenced the significance of positional encoding (PE) to achieve **high-quality** representations. Parametric PEs, such as hash grids and orthogonal planes with **learnable** parameters [A,B,C], exhibit a superior expressive capacity compared to fixed PE (e.g., Fourier feature mappings [D]).
>
>     The outstanding expressive power of parametric PEs suggests a promising direction for generative models, favoring learning to generate PE over MLP weights. In [E], they have demonstrated the generation of high-fidelity 3D-aware 2D images by generating tri-plane PEs based on GAN. Unlike the prior work, we propose PE generation with the latent **diffusion model**, enhancing it by our novel modules such as **multi-resolution framework (HDBF)** and **coarse-to-fine conditioning (CFC)**.
>
>     Indeed, at the early stage of this project, we developed a latent diffusion model for weight generation (with fixed PEs) with a hypernetwork to modulate the MLP weight of INR ($\pi$) with latent from the Encoder. The weight generation model exhibited significantly slower convergence and failed to produce high-quality samples. Due to limited resources, we are not able to provide experimental results to compare weight vs PE generation (without HDBFs and CFC). We will add the comparison in the final version.
>
>     ---
>
>
>     [A] Muller et al., Instant neural graphics primitives with a multi-resolution hash encoding, ACM ToG 2022
>
>     [B] Cao et al., Hexplane: A fast representation for dynamic scenes, CVPR 2023
>
>     [C] Fridovich-Keil et al., K-planes: Explicit radiance fields in space, time, and appearance, CVPR 2023
>
>     [D] Mildenhall et al., NeRF: Representing Scenes as Neural Radiance Fields for View Synthesis, ECCV 2020
>
>     [E] Chan et. al., Efficient geometry-aware 3d generative adversarial networks, CVPR 2022
>
> $$\\\\$$
>
> 2. **[W3] Scalability**
>
>     Thank you for your comments on scalability. We have evaluated our method on larger datasets such as CIFAR10 and LSUN church, which contain 50K and 126K samples respectively, as detailed in Table 11 of the appendix. While we acknowledge the value of scaling to even larger datasets like Objaverse, our current resources limit this expansion. Nevertheless, we see this as a crucial next step and are exploring opportunities for such large-scale evaluations.
>
> $$\\\\$$
>
> 3. **[W4] Significance of domain-agnostic architecture**
>
>     Thank you for your insightful comment. Our domain-agnostic architecture, providing a consistent latent type for LDM, enables learning the unified priors across multiple domains, as you mentioned. We believe that our approach paves the way to a large-scale, multi-modal foundation model that encompasses 2D, 3D, or even other data types. Such a model could serve as a new tool for a wide range of cross-domain downstream tasks.
>
> $$\\\\$$
>
> 4. **[W5] Domain-agnostic architecture of DDMI**
>
>     Our framework proposes a **domain-agnostic meta-architecture** for INR generation. Also, overall training and generation schemes are the same across multiple domains. This domain-agnostic meta-architecture allows us to use identical diffusion models in the latent space for generating 2D grid latent variables $\textbf{z}$.

---

> > ### Author Response · Authors · 2023-11-20
> > **Reply to Reviewer 6uL5 (2)**
> >
> > As you suggested, we provide the comparison between weight vs PE generation (without HDBFs and CFC), where the result **aligns** with our hypothesis in the paper, i.e., PE generation works better than weight generation. In detail, we trained our framework **without** HDBFs and CFC on CelebA-hq 64x64. The results, as illustrated in the table below, demonstrate that even in the absence of these two modules, our model outperforms the baselines in terms of FID, including the weight generation approach (Functa).
> >
> > |  | CelebA-HQ 64x64 |
> > | --- | --- |
> > | Functa | 40.4 |
> > | GEM | 30.4 |
> > | GASP | 13.5 |
> > | DPF | 13.2 |
> > | Ours (w/o HDBF, CFC) | 10.92 |
> > | Ours (w HDBF, CFC) | 9.74 |

---

### Meta-Review · Area_Chair_NviG · 2023-12-06

**Metareview:**

Note: unfortunately, in the current version of the paper figures are not available and prior versions are also not shown in Openreview. This makes it more difficult to understand the merit of the paper.

The paper proposes a domain-agnostic generative modeling approach based on implicit neural representations (INRs) and in particular modeling positional embeddings (PEs) instead of the weights of the INR MLP directly.The method is competitive with relevant baselines on several datasets of images, 3D shapes, and videos.

After rebuttal and discussion, the reviewers are positive about the paper. Below are main pros and cons.

Pros:
1. A reasonable and effective method
2. Fairly thorough experiments on various domains and datasets
3. Reasonable ablations and interesting analysis

Cons:
1. Somewhat limited technical innovation - most components have been proposed before
2. Experiments mostly on relatively small datasets
3. It is not entirely clear if the proposed method is indeed domain-agnostic given that it needs different parameters for different domains/datasets

All in all, the method is interesting and performs well, so the paper can be useful to the readers. Therefore, I recommend acceptance. However, I encourage the authors to address the reviewers' comments, including certainly releasing the code. And adding back the figures.

**Justification For Why Not Higher Score:**

1. Somewhat limited technical innovation - most components have been proposed before
2. Experiments mostly on relatively small datasets
3. It is not entirely clear if the proposed method is indeed domain-agnostic given that it needs different parameters for different domains/datasets

**Justification For Why Not Lower Score:**

1. A reasonable and effective method
2. Fairly thorough experiments on various domains and datasets
3. Reasonable ablations and interesting analysis

---

### Decision · Program_Chairs · 2024-01-16

Accept (poster)